

# Generalized Carter constant for quadrupolar test bodies in Kerr spacetime

Geoffrey Compère[1⋆], Adrien Druart[1†] and Justin Vines[2,3†]

**1** Université Libre de Bruxelles, International Solvay Institutes,
CP 231, B-1050 Brussels, Belgium
**2** Max Planck Institute for Gravitational Physics (Albert Einstein Institute),
Am Mülenberg 1, Potsdam 14476, Germany
**3** Mani L. Bhaumik Institute for Theoretical Physics,
Department of Physics and Astronomy, UCLA, Los Angeles, USA

⋆ geoffrey.compere@ulb.be , † Adrien.Druart@ulb.be , ‡ justin.vines@aei.mpg.de

## Abstract

We establish the existence of a deformation of the usual Carter constant which is conserved along the motion in a fixed Kerr background of a spinning test body possessing the spin-induced quadrupole coupling of a black hole. The conservation holds perturbatively up to second order in the test body's spin. This constant of motion is obtained through the explicit resolution of the conservation constraint equations, employing covariant algebraic and differential relations amongst covariant building blocks of the Kerr background. For generic spin-induced quadrupole couplings, which describe compact objects such as neutron stars, we obtain a no-go result on the existence of such a conserved quantity.

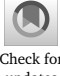

# 1 Introduction

The Kerr geometry, which describes the spacetime outside an isolated rotating black hole according to general relativity, possesses a *hidden symmetry*, not corresponding to any spacetime isometry, responsible for the integrability of geodesic motion and for the separability of various field equations. The spacetime possesses only two independent Killing vector fields, $\xi^\mu$ and $\eta^\mu$, generating the isometries of time-translation and rotation about the spin axis, leading to the energy $E = -\xi^\mu p_\mu$ and the axial component of angular momentum $L_z = \eta^\mu p_\mu$, respectively, as conserved quantities for geodesic motion with momentum $p_\mu$, i.e. $p^\nu \nabla_\nu p_\mu = 0$. It was thus rather unexpected when Carter [1] found a further nontrivial constant of the motion which is quadratic in the momentum, of the form $Q = K^{\mu\nu} p_\mu p_\nu$, where $K^{\mu\nu} = K^{\nu\mu}$ is a Killing tensor satisfying $\nabla_{(\mu} K_{\nu)\rho} = 0$, a generalization of the Killing equation $\nabla_{(\mu} \xi_{\nu)} = 0$. Adding the Carter constant $Q$ to $E$, $L_z$ and $m^2 = -g^{\mu\nu} p_\mu p_\nu$ (conserved for geodesics in any background), there are four Poisson-commuting constants of motion, ensuring that geodesic motion in the Kerr spacetime is a fully integrable dynamical system. This implies the absence of chaos in the motion, and it reduces the problem of solving for geodesic trajectories to the task of performing one-dimension integrals.

While the geodesic equation describes the motion of a structureless monopolar test body in a fixed background spacetime, an important generalization is to allow the test body (while still having negligible mass and thus negligible influence on the gravitational field) to have a finite size and nontrivial structure. In the case where such an "extended test body" has a size (length scale) $l$ which is small compared to the radius of curvature $R$ of the background, $l \ll R$, it is usefully characterized by a centroid worldline $x = z(\tau)$, with tangent $v^\mu \triangleq \mathrm{d}z^\mu/\mathrm{d}\tau$, and a tower of gravitational multipole moments defined along the worldline. These begin with the momentum $p_\mu$ as the monopole and the spin (relativistic angular momentum) tensor $S_{\mu\nu} = -S_{\nu\mu}$ as the dipole. Using only the fact that the body is described by a test stress-energy tensor $T_{\mu\nu}$ which is conserved within the background, $\nabla_\mu T^{\mu\nu} = 0$, and certain definitions of the multipole moments as spatial integrals of $T^{\mu\nu}$ (which reduce locally to standard definitions in special relativity), one can show that the monopole $p_\mu$ and dipole $S_{\mu\nu}$ must evolve along the worldline according to the Mathisson-Papapetrou-Dixon (MPD) equations [2–4],

$$\frac{\mathrm{D}p^\mu}{\mathrm{d}\tau} = -\frac{1}{2} R^\mu{}_{\nu\alpha\beta} v^\nu S^{\alpha\beta} + \dots, \tag{1a}$$

$$\frac{\mathrm{D}S^{\mu\nu}}{\mathrm{d}\tau} = 2 p^{[\mu} v^{\nu]} + \dots, \tag{1b}$$

reflecting local conservation of momentum and angular momentum, where the dots represent corrections due to the quadrupole and higher multipole moments. These are to be supplemented by a condition of the form $w^\mu S_{\mu\nu} = 0$ for some timelike vector field $w^\mu$ (setting to zero the mass-dipole moment in the frame of $w^\mu$), which fixes a choice of centroid worldline; a convenient choice is the Tulczyjew-Dixon condition $p^\mu S_{\mu\nu} = 0$ [5,6]. Along with such a condition, the pole-dipole MPD equations (Eqs. (1) with the dots dropped, neglecting the quadrupole and higher corrections) provide a closed set of evolution equations for the worldline $z(\tau)$ and the momentum $p^\mu(\tau)$ and spin $S^{\mu\nu}(\tau)$ along it, describing the motion of spinning test body a background curved spacetime.

As shown by Dixon [4,7], and as was central to his construction of the multipole moments, if the background has a Killing vector $\xi^\mu$, then the quantity

$$\mathcal{P}_\xi = p_\mu \xi^\mu + \frac{1}{2} S^{\mu\nu} \nabla_\mu \xi_\nu, \tag{2}$$

is exactly conserved along any worldline when $p^\mu$ and $S^{\mu\nu}$ are evolved by the MPD equations (1), to all orders in the multipole expansion, for arbitrary quadrupole and higher moments. (See also, e.g., the earlier derivation by Souriau for the pole-dipole system [8], and the insightful exposition by Harte [9].) Considering a background Kerr spacetime, one is then naturally lead to wonder whether the hidden symmetry leads to conserved quantities for the MPD dynamics, including a generalization of the Carter constant to the case of spinning extended test bodies.

This question was answered for the case of the pole-dipole MPD equations by Rüdiger [10,11]. First he showed that the quantity $Q_Y = {}^*Y^{\mu\nu} S_{\mu\nu}$ built from the Killing-Yano tensor $Y_{\mu\nu}$ of Kerr is conserved, up to remainders quadratic in the spin tensor and quadrupolar corrections. He further showed that there is indeed a generalized Carter constant of the form

$$Q^{(2)} = K^{\mu\nu} p_\mu p_\nu + L^{\mu\nu\rho} S_{\mu\nu} p_\rho + \dots, \tag{3}$$

which is conserved along (1) with $p^\mu S_{\mu\nu} = 0$, up to remainders quadratic in the spin tensor and quadrupolar corrections that were not determined. A generalization of this result to the Kerr-Newman (charged spinning black hole) spacetime was independently discovered by Gibbons *et al.* [12] using a supersymmetric description of spinning particle dynamics. The existence of these constants of motion has been shown by Witzany to allow the separation of a

Hamilton-Jacobi equation for the pole-dipole system in Kerr, leading to analytic expressions for the fundamental frequencies of the motion [13] using a Hamiltonian formalism for spinning particles [14].

Our purpose in this paper is to explore whether such "hidden constants" exist for test bodies with spin-induced quadrupole moments moving in a Kerr background. Dixon's generalizations of the equations of motion (1) to the quadrupolar order in the test body's multipole expansion read

$$\frac{\mathrm{D}p^{\mu}}{\mathrm{d}\tau} = -\frac{1}{2}R^{\mu}{}_{\nu\alpha\beta}v^{\nu}S^{\alpha\beta} - \frac{1}{6}J^{\alpha\beta\gamma\delta}\nabla^{\mu}R_{\alpha\beta\gamma\delta} + \dots, \tag{4a}$$

$$\frac{\mathrm{D}S^{\mu\nu}}{\mathrm{d}\tau} = 2p^{[\mu}v^{\nu]} + \frac{4}{3}R^{[\mu}{}_{\alpha\beta\gamma}J^{\nu]\alpha\beta\gamma} + \dots, \tag{4b}$$

where $J^{\mu\nu\rho\sigma}(\tau)$ is the quadrupole tensor, having the same symmetries as the Riemann tensor, and the ellipses here represent octupolar and higher corrections. As has been developed and applied in a number of works (see e.g. [15–23]), the form of $J$ appropriate to describe a spin-induced quadrupole moment is given by

$$J^{\mu\nu\rho\sigma} = \kappa\frac{3p\cdot v}{(p^2)^2}p^{[\mu}S^{\nu]\lambda}S^{[\rho}{}_{\lambda}p^{\sigma]}, \tag{5}$$

where $\kappa$ is a response coefficient controlling the magnitude of the quadrupolar deformation, proportional to the square of the spin. Typical values for $\kappa$ for a neutron star are in the range 4 to 8 [24], while for a black hole $\kappa_{\mathrm{BH}} = 1$.

As the central results of this paper, we establish that two quantities, $Q_Y$ and $Q^{(2)}$, are conserved up to cubic-in-spin or octupolar corrections,

$$\frac{\mathrm{d}Q_Y}{\mathrm{d}\tau} = \mathcal{O}(\mathcal{S}^3), \qquad \frac{\mathrm{d}Q^{(2)}}{\mathrm{d}\tau} = \mathcal{O}(\mathcal{S}^3), \tag{6}$$

along the motion of a "quadrupolar test black hole", governed by (4)–(5) with $\kappa = 1$ and with the Tulczyjew condition $p_{\mu}S^{\mu\nu} = 0$, in a background Kerr spacetime, for arbitrary orbital and spin orientations. The first quantity $Q_Y$ is Rüdiger's linear-in-spin constant, unmodified,

$$Q_Y = {}^*Y^{\mu\nu}S_{\mu\nu}, \tag{7}$$

where ${}^*Y^{\mu\nu} = \frac{1}{2}\varepsilon^{\mu\nu\alpha\beta}Y_{\alpha\beta}$ is the dual of the Kerr spacetime's nontrivial Killing-Yano tensor $Y^{\mu\nu}$. The second quantity $Q^{(2)}$ is quadratic in $p$ and $S$ and generalizes Rüdiger's constant (3) to the quadrupolar order for a test black hole; it is given explicitly by

$$Q^{(2)} = Y_{\mu\rho}Y^{\rho}{}_{\nu}p^{\mu}p^{\nu} + 4\xi^{\lambda}\varepsilon_{\lambda\mu\sigma[\rho}Y_{\nu]}{}^{\sigma}S^{\mu\nu}p^{\rho}$$
$$-\left[g_{\mu\rho}\left(\xi_{\nu}\xi_{\sigma} - \frac{1}{2}g_{\nu\sigma}\xi^2\right) - \frac{1}{2}Y_{\mu}{}^{\lambda}\left(Y_{\rho}{}^{\kappa}R_{\lambda\nu\kappa\sigma} + \frac{1}{2}Y_{\lambda}{}^{\kappa}R_{\kappa\nu\rho\sigma}\right)\right]S^{\mu\nu}S^{\rho\sigma}, \tag{8}$$

where $\xi^{\mu}$ is the timelike Killing vector, and it reduces to the Carter constant $K_{\mu\nu}p^{\mu}p^{\nu} = Y_{\mu\rho}Y^{\rho}{}_{\nu}p^{\mu}p^{\nu}$ for geodesic motion when the test body's spin is set to zero.

After reviewing details of the motion of quadrupolar test bodies in curved spacetime in Sec. 2, we develop the constraint equations for the existence of the conserved quantities in tensorial form in Sec. 3. In Sec. 4, we discuss covariant algebraic and differential relations amongst basic fields, "covariant building blocks," characterizing the Kerr geometry, which play a central role in our solutions to the constraint equations. We use these to reduce the tensorial constraint equations to a system of scalar equations in Sec. 5, and we derive our solutions for the special black-hole case $\kappa = 1$ in Sec. 6. In Sec. 7, we investigate the case $\kappa \neq 1$, for non-black-hole bodies such as neutron stars with spin-induced quadrupoles, concluding that

there is no solution to the constraints for $\kappa \neq 1$. Finally we summarize our findings, some aspects of their broader context, and future directions in Sec. 8.

We use the same conventions as adopted previously in [25]. The main results of this paper have been crosschecked using *Mathematica*, which was also used to performed the numerical evaluations of Section 7. The various notebooks can be found on the GitHub repository https://github.com/addruart/generalizedCarterConstant.

# 2 Quadrupolar test bodies in curved spacetime

In this section we review the motion of test-bodies in curved spacetime and we discuss the problem of finding conserved quantities associated to the corresponding dynamical system.

## 2.1 Motion of test bodies in curved spacetime

**Evolution equations.** In General Relativity, the motion of an extended test body over a curved background is described by the Mathisson-Papapetrou-Dixon (MPD) equations [2–4]:

$$\frac{\mathrm{D}p^{\mu}}{\mathrm{d}\tau} = -\frac{1}{2}R^{\mu}{}_{\nu\alpha\beta}v^{\nu}S^{\alpha\beta} + \mathcal{F}^{\mu}, \tag{9a}$$

$$\frac{\mathrm{D}S^{\mu\nu}}{\mathrm{d}\tau} = 2p^{[\mu}v^{\nu]} + \mathcal{L}^{\mu\nu}. \tag{9b}$$

Here, $v^{\mu}$ denotes the four-velocity of the test body, $p^{\mu}$ its four-impulsion and $S^{\mu\nu}$ its spin dipole tensor. $\mathcal{F}^{\mu}$ and $\mathcal{L}^{\mu\nu}$ are respectively the force and torque terms that include corrections to the equations of motion due to quadrupole and higher moments. From now on, we assume that $\tau$ is the proper time, yielding $v_{\mu}v^{\mu} = -1$.

Taken alone, the MPD equations do not form a closed set of equations. Roughly speaking, this comes from the fact that the MPD equations arise from the skeletonization of a compact body stress-energy tensor above an arbitrary worldline belonging to the body's worldtube [2]. They shall be supplemented by a so-called *spin supplementary condition* (SSC), which will play the role of fixing this worldline [26]. In this paper, we choose to work with Tulczyjew SSC [5,6]

$$p_{\mu}S^{\mu\nu} = 0. \tag{10}$$

Among other consequences extensively described in [25], enforcing this SSC allows in particular to express the dipole spin tensor solely in terms of a spin vector $S^{\mu}$ which is orthogonal to the impulsion, $S_{\mu}p^{\mu} = 0$.

**Spin-induced quadrupole approximation.** We will consider only spin-induced multipole moments and work in the quadrupole approximation, *i.e.* neglecting octupole and higher moments. This is the relevant approximation for addressing spin-squared interactions: considering only spin-induced multipole terms, the $2^{n}$-pole scales as $\mathcal{O}(\mathcal{S}^{n})$, with $\mathcal{S}^{2} \triangleq \frac{1}{2}S_{\mu\nu}S^{\mu\nu}$.

At the level of the equations of motion, this corresponds to choose the force and torque given by

$$\mathcal{F}^{\mu} = -\frac{1}{6}J^{\alpha\beta\gamma\delta}\nabla^{\mu}R_{\alpha\beta\gamma\delta}, \qquad \mathcal{L}^{\mu\nu} = \frac{4}{3}R^{[\mu}{}_{\alpha\beta\gamma}J^{\nu]\alpha\beta\gamma}. \tag{11}$$

The quadrupole tensor $J^{\mu\nu\rho\sigma}$ possesses the same algebraic symmetries as the Riemann tensor.

We further particularize our study by considering only a quadrupole moment that is induced by the spin of the body, discarding the possible presence of some intrinsic quadrupole

moment. This *spin-induced quadrupole* can be shown to take the form (5) [15–23]. Here specialized to the case $v_\alpha v^\alpha = -1$ and at leading order in the spin expansion using Eq. (19), its form reduces to

$$J^{\mu\nu\rho\sigma} = \frac{3\kappa}{\mu} v^{[\mu} S^{\nu]\lambda} S_\lambda{}^{[\rho} v^{\sigma]} = -\frac{3\kappa}{\mu} v^{[\mu} \Theta^{\nu][\rho} v^{\sigma]}, \qquad \text{where } \Theta^{\alpha\beta} \triangleq S^{\alpha\lambda} S^\beta{}_\lambda. \tag{12}$$

Here $\kappa$ is a free coupling parameter that equals 1 for a Kerr black hole and takes another value if the test-body is another compact object, *e.g.* a neutron star.

Under the Tulczyjew SSC, the spin tensor can be solely expressed in terms of a spin vector defined as $S^\alpha \triangleq \frac{1}{2}\varepsilon^{\alpha\beta\gamma\delta}\hat{p}_\beta S_{\gamma\delta}$. This relation can be inverted as

$$S^{\alpha\beta} = 2S^{[\alpha}\hat{p}^{\beta]*}, \tag{13}$$

were we have introduced the Hodge duality $A^*_{\mu\nu} \triangleq \frac{1}{2}\varepsilon_{\mu\nu\rho\sigma}A^{\rho\sigma}$ (which is here specialized to the outer product of vectors, $l^{[\mu}m^{\nu]*} \triangleq \frac{1}{2}\varepsilon^{\mu\nu\rho\sigma}l_\rho m_\sigma$). One can show that this implies the following decomposition for $\Theta^{\alpha\beta}$:

$$\Theta^{\alpha\beta} = \Pi^{\alpha\beta}\mathcal{S}^2 - S^\alpha S^\beta, \tag{14}$$

with $\Pi^\alpha_\beta \triangleq \delta^\alpha_\beta + \hat{p}^\alpha \hat{p}_\beta$ the projector on the hypersurface orthogonal to the timelike unit vector $\hat{p}^\alpha \triangleq \frac{p^\alpha}{\sqrt{-p_\alpha p^\alpha}}$ and $\mathcal{S}^2 = \frac{1}{2}S_{\alpha\beta}S^{\alpha\beta} = S_\alpha S^\alpha$. Moreover, one has the identities

$$\mathcal{F}^\mu = \frac{\kappa}{2\mu}\hat{p}^\alpha \Theta^{\beta\gamma}\hat{p}^\delta \nabla^\mu R_{\alpha\beta\gamma\delta} + \mathcal{O}(\mathcal{S}^3), \tag{15a}$$

$$\mathcal{L}^{\mu\nu} = \frac{2\kappa}{\mu}R^\nu{}_{\alpha\beta\gamma}v^{[\mu}\Theta^{\alpha]\beta}v^\gamma - (\mu \leftrightarrow \nu), \tag{15b}$$

$$\mathcal{L}^{\mu\nu}v_\nu = \frac{\kappa}{\mu}\Big(\hat{p}^\mu \hat{p}^\nu R_{\nu\alpha\beta\gamma} + R^\mu{}_{\alpha\beta\gamma}\Big)\Theta^{\alpha\beta}\hat{p}^\gamma + \mathcal{O}(\mathcal{S}^4) = \frac{\kappa}{\mu}\Pi^{\mu\nu}R_{\nu\alpha\beta\gamma}\Theta^{\alpha\beta}\hat{p}^\gamma + \mathcal{O}(\mathcal{S}^4). \tag{15c}$$

**Conservation of the spin, mass; relation between four-velocity and impulsion.** We define the invariant and kinematic masses, respectively given by

$$\mu^2 \triangleq -p_\alpha p^\alpha, \qquad \mathfrak{m} \triangleq -v_\alpha p^\alpha. \tag{16}$$

Differentiating the SSC (10) yields

$$\mu^2 v^\mu - \mathfrak{m}p^\mu = \frac{1}{2}S^{\mu\nu}R_{\nu\lambda\rho\sigma}v^\lambda S^{\rho\sigma} - \mathcal{L}^{\mu\nu}p_\nu - S^{\mu\nu}\mathcal{F}_\nu. \tag{17}$$

Contracting this equation with $v_\mu$ provides us with

$$\mu^2 = \mathfrak{m}^2 + \mathcal{O}(\mathcal{S}^3). \tag{18}$$

The expression of the 4-impulsion in terms of the 4-velocity reads as

$$p^\mu = \mu v^\mu - \frac{1}{2\mu}S^{\mu\nu}R_{\nu\lambda\rho\sigma}v^\lambda S^{\rho\sigma} + \mathcal{L}^{\mu\nu}v_\nu + \mathcal{O}(\mathcal{S}^3). \tag{19}$$

In the quadrupole approximation, $\mu$ is no longer conserved at $\mathcal{O}(\mathcal{S}^2)$, since

$$\frac{\mathrm{d}\mu}{\mathrm{d}\tau} = -v_\mu \mathcal{F}^\mu + \mathcal{O}(\mathcal{S}^3). \tag{20}$$

However, notice that, provided we assume[1]

$$\frac{\mathrm{D}}{\mathrm{d}\tau} J^{\alpha\beta\gamma\delta} = \mathcal{O}(\mathcal{S}^3), \tag{21}$$

one can still define a mass-like quantity, given by

$$\tilde{\mu} \triangleq \mu - \frac{1}{6} J^{\alpha\beta\gamma\delta} R_{\alpha\beta\gamma\delta}, \tag{22}$$

which is quasi-conserved, namely

$$\frac{\mathrm{d}\tilde{\mu}}{\mathrm{d}\tau} = \mathcal{O}(\mathcal{S}^3). \tag{23}$$

Moreover, one can perturbatively invert (19) to obtain an expression of the four-velocity in terms of the impulsion and the spin:

$$v^\mu = \hat{p}^\mu + (D^{\mu\nu} - \frac{1}{\mu}\mathcal{L}^{\mu\nu})\hat{p}_\nu + \mathcal{O}(\mathcal{S}^3), \tag{24}$$

with

$$\hat{p}^\alpha \triangleq \frac{p^\alpha}{\mu} = \frac{p^\alpha}{\tilde{\mu}} + \mathcal{O}(\mathcal{S}^2), \qquad D^\mu{}_\nu \triangleq \frac{1}{2\mu^2} S^{\mu\lambda} R_{\lambda\nu\rho\sigma} S^{\rho\sigma}. \tag{25}$$

Eq. (24) will play a central role when we will work out the conservation equations in the following sections.

The spin magnitude $\mathcal{S}^2$ is exactly conserved [27]

$$\frac{d}{d\tau}(\mathcal{S}^2) = 0. \tag{26}$$

## 2.2 Searching for conserved quantities: Rüdiger's procedure

In two papers published in the early 80s, Rüdiger described a scheme for constructing quantities conserved along the motion driven by the MPD equations [10, 11]. The basic guideline followed in his scheme was to enforce directly the conservation equation on a generic Ansatz for the conserved quantity, and to subsequently solve the constraints obtained. This procedure was extensively reviewed and discussed in [25]. We provide here a short summary of its main steps, which would allow the reader to get familiar with our terminology and notations.

⋄ **Step 1: postulate an Ansatz for the conserved quantity.** The conserved quantity should be a function of the dynamical variables $p^\mu$ and $S^\mu$. It is therefore a function $Q(x^\mu, S^\mu, p^\mu)$. Assuming its analyticity, it can be expanded as

$$Q(x^\mu, S^\alpha, p^\mu) = \sum_{\substack{s,p \geq 0 \\ s+p > 0}} Q^{[s,p]}(x^\mu, S^\alpha, p^\mu), \tag{27}$$

with

$$Q^{[s,p]}(x^\mu, S^\alpha, p^\mu) \triangleq Q^{[s,p]}_{\alpha_1...\alpha_s\mu_1...\mu_p}(x^\mu) S^{\alpha_1}...S^{\alpha_s} p^{\mu_1}...p^{\mu_p}. \tag{28}$$

Expressions like this one – that is, tensorial quantities fully contracted with occurrences of the impulsion and the spin – will often appear in the following computations. It is useful to enable a distinction between them by introducing a grading allowing the counting of the number of occurrences of both the spin vector $S^\mu$ and the impulsion $p^\mu$, which is provided by the notation $[s, p]$. More generally, we define:

---

[1]This condition is automatically satisfied for the spin-induced quadrupole.

**Definition 1.** *A fully-contracted expression of the type*

$$T_{\alpha_1 \dots \alpha_s \mu_1 \dots \mu_p} \ell_s^{\alpha_1} \dots \ell_s^{\alpha_s} \ell_p^{\mu_1} \dots \ell_p^{\mu_p}, \tag{29}$$

*where* $\ell_s^{\alpha} = S^{\alpha}, s^{\alpha}$ *(the relaxed spin vector* $s^{\mu}$ *will be defined below) and* $\ell_p^{\mu} = p^{\mu}, \hat{p}^{\mu}$ *is said to be of grading* $[s, p]$. *Equivalently,* $s$ *(resp.* $p$*) will be referred to as the spin (resp. momentum) grading of this expression.*

Since we have only included the quadrupole term in the equations of motion but neglected all the $\mathcal{O}(\mathcal{S}^3)$ terms, it is not self-consistent to look at quantities which are conserved beyond second order in the spin magnitude. We therefore restrict our analysis to Ansätze that contain terms of of spin grading at most equal to two. Historically, Rüdiger didn't consider the full set of possible Ansätze originating from this discussion, but only the two restricted cases

$$Q^{(1)} \triangleq \sum_{p=1} Q^{[s,p]} \triangleq X_{\mu} p^{\mu} + W_{\mu\nu} S^{\mu\nu}, \tag{30}$$

$$Q^{(2)} \triangleq \sum_{p=2} Q^{[s,p]} \triangleq K_{\mu\nu} p^{\mu} p^{\nu} + L_{\mu\nu\rho} S^{\mu\nu} p^{\rho} + M_{\mu\nu\rho\sigma} S^{\mu\nu} S^{\rho\sigma}. \tag{31}$$

We will refer to them as respectively the *linear* and the *quadratic* invariants in $p^{\mu}$. They are homogeneous in the number of occurrences of $p^{\mu}$ and $S^{\mu\nu}$ they contain. As long as we consider the MPD equations at linear order in the spin magnitude or at quadratic order with the quadrupole coupling of the test body being the one of a black hole ($\kappa = 1$), it turns out that considering only these two types of ansatzes will be enough to derive a complete set of conserved quantities. However, a more general ansatz will be necessary to consider arbitrary quadrupole couplings ($\kappa \neq 1$), as discussed in Section 7.

◇ **Step 2: write down the conservation equation.** We only require our quantity to be conserved up to second order in the spin magnitude,

$$\dot{Q} \triangleq v^{\lambda} \nabla_{\lambda} Q \overset{!}{=} \mathcal{O}(\mathcal{S}^3). \tag{32}$$

◇ **Step 3: expand the conservation equation using the equations of motion.** The next step is to plug the explicit form of the Ansatz chosen in the conservation equation, and to use the MPD equations (9) to replace the covariant derivatives of the impulsion and of the spin tensor. The occurrences of the four-velocity are replaced by the means of Eq. (24).

◇ **Step 4: express the conservation equation in terms of independent variables.** The presence of a SSC make the variables $p^{\mu}$, $S^{\alpha\beta}$ not independent among themselves. We turn to an independent set of variables in two steps: (i) we use the relation $S^{\alpha\beta} = 2S^{[\alpha} \hat{p}^{\beta]*}$ to replace all the spin tensors $S^{\mu\nu}$ by the spin vectors $S^{\mu}$ and (ii) we replace the occurrences of the spin vector by the *relaxed spin vector* $s^{\alpha}$ defined through

$$S^{\alpha} = \Pi^{\alpha}_{\beta} s^{\beta}. \tag{33}$$

It allows to relax the residual constraint $S_{\mu} p^{\mu} = 0$ by considering a spin vector possessing a non-vanishing component along the direction of the impulsion. Physical quantities will be independent of this component. It is introduced in order to decouple the conservation equation. For convenience, we scale the unphysical component of the relaxed spin vector such that $s_{\alpha} s^{\alpha} \sim S_{\alpha} S^{\alpha} = \mathcal{S}^2$. Notice that we have the useful identity

$$S^{[\alpha} p^{\beta]} = s^{[\alpha} p^{\beta]} \quad \Rightarrow \quad S^{\alpha\beta} = 2s^{[\alpha} \hat{p}^{\beta]*}. \tag{34}$$

⬦ **Step 5: infer the independent constraints.** The conservation equation takes now the form of a sum of fully-contracted expressions of the type (29), involving only the *independent* dynamical variables $p^\mu$ and $s^\alpha$:

$$\dot{Q} = \sum_{\substack{s,p \geq 0 \\ s+p>0}} T^{[s,p]}_{\alpha_1 \dots \alpha_s \mu_1 \dots \mu_p} s^{\alpha_1} \dots s^{\alpha_s} p^{\mu_1} \dots p^{\mu_p} \overset{!}{=} \mathcal{O}(\mathcal{S}^3). \tag{35}$$

The conservation equation is then equivalent to the requirement that all the terms of different gradings $[s,p]$ vanish independently:

$$T^{[s,p]}_{\alpha_1 \dots \alpha_s \mu_1 \dots \mu_p} s^{\alpha_1} \dots s^{\alpha_s} p^{\mu_1} \dots p^{\mu_p} \overset{!}{=} \mathcal{O}(\mathcal{S}^3). \tag{36}$$

$s^\alpha$ and $p^\mu$ being arbitrary, this is equivalent to the *constraint equations*

$$T^{[s,p]}_{(\alpha_1 \dots \alpha_s)(\mu_1 \dots \mu_p)} \overset{!}{=} \mathcal{O}(\mathcal{S}^3). \tag{37}$$

⬦ **Step 6: find a solution and prove uniqueness.** This final step is non-systematic. For the simplest cases (linear invariant with black hole quadrupole coupling, quadratic invariant at first order in the spin magnitude), it will be sufficient to work only with the tensorial constraints (37). However, for more involved cases (linear invariant with arbitrary quadrupole coupling, quadratic invariant at second order in the spin), the tensorial relations will become so cumbersome that turning to another formulation of the problem will appear to be fruitful. This will be the purpose of the covariant building blocks for Kerr introduced in Section 4.

## 3 Constraint equations: Tensorial formulation

In this section, we will apply the aforementioned procedure to derive the tensorial constraint equations that should be obeyed for ensuring the conservation of the two quantities (30), (31).

### 3.1 Linear constraint

Following Rüdiger, we start from the Ansatz (30) for the linear invariant:

$$Q^{(1)} \triangleq X_\mu p^\mu + W_{\mu\nu} S^{\mu\nu}. \tag{38}$$

Notice that $W_{\mu\nu}$ should be a skew-symmetric tensor. The time variation of (30) is given by

$$\dot{Q}^{(1)} = v^\lambda \big( \nabla_\lambda X_\mu p^\mu + X_\mu \nabla_\lambda p^\mu + \nabla_\lambda W_{\mu\nu} S^{\mu\nu} + W_{\mu\nu} \nabla_\lambda S^{\mu\nu} \big). \tag{39}$$

Applying Rüdiger's procedure, the conservation equation $\dot{Q}^{(1)} = \mathcal{O}(\mathcal{S}^3)$ reduces to the following set of equations:

$$[0,2]: \qquad \nabla_\mu X_\nu \hat{p}^\mu \hat{p}^\nu = \mathcal{O}(\mathcal{S}^3), \tag{40a}$$

$$[1,2]: \qquad \nabla_\mu Y_{\alpha\nu} s^\alpha \hat{p}^\mu \hat{p}^\nu - \frac{1}{2} X^\lambda R^*_{\lambda\nu\beta\rho} s^\beta \hat{p}^\nu \hat{p}^\rho = \mathcal{O}(\mathcal{S}^3), \tag{40b}$$

$$[2,2]: \qquad \frac{\kappa}{2\mu} X^\lambda \nabla_\lambda R_{\nu\alpha\beta\rho} s^\alpha s^\beta \hat{p}^\nu \hat{p}^\rho + Y_{\mu\nu} \mathcal{L}^{*\mu\nu} = \mathcal{O}(\mathcal{S}^3), \tag{40c}$$

$$[2,4]: \qquad \big( \nabla_\lambda X_\mu - 2W_{\lambda\mu} \big) \big( \mu D^\lambda{}_\nu - \mathcal{L}^\lambda{}_\nu \big) \hat{p}^\mu \hat{p}^\nu = \mathcal{O}(\mathcal{S}^3). \tag{40d}$$

Here, we have introduced the notation $Y_{\mu\nu} \triangleq W^*_{\mu\nu}$. We therefore have the following proposition:

**Proposition 1.** *For any pair $(X_\mu, W_{\mu\nu})$ satisfying the constraint equations (40) and assuming the MPD equations (9) are obeyed, the quantity $Q^{(1)}$ (39) will be conserved up to second order in the spin parameter, i.e. $\dot{Q}^{(1)} = \mathcal{O}(\mathcal{S}^3)$.*

Two independent classes of solutions to these constraint equations can be constructed:

⋄ For $X_\mu \neq 0$, the $[0,2]$ equation (40a) requires that $X_\mu$ should be a Killing vector field,

$$\nabla_{(\mu} X_{\nu)} = 0 \,. \tag{41}$$

In this case, making use of the Kostant formula

$$\nabla_\alpha \nabla_\beta X_\mu = R_{\mu\nu\alpha\beta} X^\nu \,, \tag{42}$$

which holds for any Killing vector $X_\mu$, the $[1,2]$ constraint (40b) reduces to

$$\nabla_\mu (W_{\alpha\beta} - 2\nabla_\alpha X_\beta) \hat{p}^\mu S^{\alpha\beta} = \mathcal{O}(\mathcal{S}^3) \,. \tag{43}$$

It is clear that this constraint as well as the $[2,4]$ constraint (40d) are solved by

$$W_{\alpha\beta} = \frac{1}{2} \nabla_\alpha X_\beta \,. \tag{44}$$

A little more work is necessary to show that the remaining constraint Eq. (40c) also holds for this value of $Y_{\alpha\beta}$. At the end of the day, we have recovered the well-known conservation of the quantity

$$Q^{(1)} = X_\mu p^\mu + \frac{1}{2} \nabla_\mu X_\nu S^{\mu\nu} \,. \tag{45}$$

The conservation can be shown to be exact and to hold at any order of the multipolar expansion [7].

⋄ A second, independent solution may be obtained by considering $X_\mu = 0$. In this case, the constraint equations (40) reduce to

$$[1,2]: \qquad \nabla_\mu Y_{\alpha\nu} S^\alpha \hat{p}^\mu \hat{p}^\nu = \mathcal{O}(\mathcal{S}^3) \,, \tag{46a}$$

$$[2,2]: \qquad Y_{\mu\nu} \mathcal{L}^{*\mu\nu} = \mathcal{O}(\mathcal{S}^3) \,, \tag{46b}$$

$$[2,4]: \qquad W_{\lambda\mu}(\mu D^\lambda{}_\nu - \mathcal{L}^\lambda{}_\nu) \hat{p}^\mu \hat{p}^\nu = \mathcal{O}(\mathcal{S}^3) \,. \tag{46c}$$

Eq. (46a) enforces $Y_{\mu\nu}$ to be a Killing-Yano tensor, up to second order corrections in the spin parameter:

$$\nabla_{(\mu} Y_{\nu)\alpha} = \mathcal{O}(\mathcal{S}^2) \,. \tag{47}$$

We consequently recover Rüdiger's linear invariant [10],

$$Q_Y = Y^*_{\alpha\beta} S^{\alpha\beta} \,, \tag{48}$$

which is well-known to be conserved at linear order in the spin magnitude. The conservation at second order assuming induced quadrupoles will be discussed in Section 5.1.

## 3.2 Quadratic constraint

It is useful to decompose the quadratic quantity (31) as

$$Q^{(2)} = Q^{\text{lin}} + Q^{\text{quad}} \,, \tag{49}$$

where

$$Q^{\text{lin}} \triangleq K_{\mu\nu} p^\mu p^\nu + L_{\mu\nu\rho} S^{\mu\nu} p^\rho \,, \qquad Q^{\text{quad}} \triangleq M_{\alpha\beta\gamma\delta} S^{\alpha\beta} S^{\gamma\delta} \,. \tag{50}$$

Here, the tensors satisfy the following identities

$$K_{\mu\nu} = K_{(\mu\nu)}\,, \qquad L_{\mu\nu\rho} = L_{[\mu\nu]\rho}\,, \qquad M_{\alpha\beta\gamma\delta} = M_{[\alpha\beta]\gamma\delta} = M_{\alpha\beta[\gamma\delta]} = M_{\gamma\delta\alpha\beta}\,. \tag{51}$$

The linear-in-spin quantity $Q^{\text{lin}}$ has been extensively studied in [25]. At linear order in the spin magnitude, the variation of $Q^{(2)}$ and the variation of $Q^{\text{lin}}$ coincide. It was shown to be given by

$$\dot{Q}^{\text{lin}} = \dot{Q}^{(2)} = \mu^3 U_{\mu\nu\rho} \hat{p}^\mu \hat{p}^\nu \hat{p}^\rho + 2\mu^{2\,\star} V_{\alpha\mu\nu\rho} s^\alpha \hat{p}^\mu \hat{p}^\nu \hat{p}^\rho + \mathcal{O}(\mathcal{S}^2)\,. \tag{52}$$

The explicit expressions of the tensors $U_{\mu\nu\rho}$ and $V_{\alpha\mu\nu\rho\sigma}$ can be found in [25]. The conservation conditions at zeroth and first order $U_{(\mu\nu\rho)} = 0$, $^\star V^\alpha{}_{(\mu\nu\rho)} = 0$ are unchanged by the presence of quadrupolar terms in the MPD equations (9). In [25], we showed that, at first order in the spin parameter, the only non-trivial stationary and axisymmetric solution to these equations above a Kerr background was Rüdiger's quadratic quasi-invariant, that will be referred to as $Q_R$. This solution corresponds to

$$K_{\mu\nu} = Y_{\mu\lambda} Y^\lambda{}_\nu\,, \quad L_{\alpha\beta\gamma} = \frac{2}{3} \nabla_{[\alpha} K_{\beta]\gamma} + \frac{4}{3} \varepsilon_{\alpha\beta\gamma\delta} \nabla^\delta \mathcal{Z}\,, \tag{53}$$

where $Y_{\alpha\beta}$ is Kerr's Killing-Yano tensor and where we have defined the scalar $\mathcal{Z} \triangleq \frac{1}{4} Y^*_{\alpha\beta} Y^{\alpha\beta}$. We can compactly write $L_{\mu\nu\rho} S^{\mu\nu} p^\rho = \left( \varepsilon_{\mu\nu\rho\sigma} \xi_\lambda Y^{\lambda\sigma} + {}^\star Y_{\mu\nu} \xi_\rho \right) S^{\mu\nu} p^\rho = 4\xi^\lambda \varepsilon_{\lambda\mu\sigma[\rho} Y_{\nu]}{}^\sigma S^{\mu\nu} p^\rho$.

From this point, we will assume that the zeroth and first orders in the spin magnitude are solved by Rüdiger's solution (53), that is, we will always set $Q^{\text{lin}} = Q_R$. This completely cancels the $\mathcal{O}(\mathcal{S}^0)$ and $\mathcal{O}(\mathcal{S}^1)$ terms in the constraint equations. The presence of quadrupole terms in the MPD equations (9) will only appear at quadratic order in $\mathcal{S}$. Hence, we are left with only one constraint, which is of grading [2, 3]. The derivation of this quadratic constraint is too long to be provided in the main text and can be found instead in Appendix A. We have now demonstrated the following proposition:

**Proposition 2.** *Any tensor $N_{\alpha\beta\gamma\delta}$ possessing the same algebraic symmetries as the Riemann tensor and satisfying the constraint equation*

$$\begin{aligned} \Big[ 4\nabla_\mu N_{\alpha\nu\beta\rho} + 2\kappa \nabla_{[\alpha} \mathcal{M}^{(1)}_{|\mu|\nu]\beta\rho} + \kappa \left( g_{\alpha\mu} Y_{\lambda\nu} - g_{\mu\nu} Y_{\lambda\alpha} \right) \xi_\kappa \,^*R^{\lambda\kappa}{}_{\beta\rho} \\ + \left( 2\kappa Y_{\alpha\mu} \xi_\lambda + (2-\kappa)\left( Y_{\lambda\mu}\xi_\alpha + Y_{\alpha\lambda}\xi_\mu \right) + 3\kappa g_{\alpha\mu} \nabla_\lambda \mathcal{Z} \right) {}^*R^\lambda{}_{\nu\beta\rho} - 3\kappa g_{\mu\nu} \nabla_\lambda \mathcal{Z} \,^*R^\lambda{}_{\alpha\beta\rho} \\ + (3\kappa - 2) \nabla_\mu \mathcal{Z} R^*_{\nu\alpha\beta\rho} \Big] s^\alpha s^\beta \hat{p}^\mu \hat{p}^\nu \hat{p}^\rho \overset{!}{=} \mathcal{O}(\mathcal{S}^3)\,, \end{aligned} \tag{54}$$

*where[2]*

$$\mathcal{M}^{(1)}_{\alpha\beta\gamma\delta} \triangleq K_{\alpha\lambda} R^\lambda{}_{\beta\gamma\delta}\,, \tag{55}$$

*gives rise to a quantity*

$$\mathcal{Q}^{(2)} = Q_R + M_{\alpha\beta\gamma\delta} S^{\alpha\beta} S^{\gamma\delta}\,, \qquad M_{\alpha\beta\gamma\delta} \triangleq {}^*N^*_{\alpha\beta\gamma\delta}\,, \tag{56}$$

*which is conserved up to second order in the spin parameter for the MPD equations with spin-induced quadrupole (9), i.e. $\dot{\mathcal{Q}}^{(2)} = \mathcal{O}(\mathcal{S}^3)$.*

---

[2]This notation $\mathcal{M}^{(1)}_{\alpha\beta\gamma\delta}$ will become clearer later on.

Our next goal with be to find a way to disentangle the $\kappa = 1$ and the $\kappa \neq 1$ problems. Without loss of generality, we set

$$N_{\alpha\beta\gamma\delta} = \triangleq N^{\mathrm{BH}}_{\alpha\beta\gamma\delta} + (\kappa - 1)N^{\mathrm{NS}}_{\alpha\beta\gamma\delta}. \tag{57}$$

Because $\kappa$ is *a priori* arbitrary, the constraint (54) turns out to be equivalent to the two independent equations

$$\Big[4\nabla_\mu N^{\mathrm{BH}}_{\alpha\nu\beta\rho} + 2\nabla_{[\alpha}\mathcal{M}^{(1)}_{|\mu|\nu]\beta\rho} + \big(g_{\alpha\mu}Y_{\lambda\nu} - g_{\mu\nu}Y_{\lambda\alpha}\big)\xi_\kappa \,{}^*R^{\lambda\kappa}{}_{\beta\rho} + \big(2Y_{\alpha\mu}\xi_\lambda + \big(Y_{\lambda\mu}\xi_\alpha + Y_{\alpha\lambda}\xi_\mu\big)$$

$$+ 3g_{\alpha\mu}\nabla_\lambda\mathcal{Z}\big)\,{}^*R^\lambda{}_{\nu\beta\rho} - 3g_{\mu\nu}\nabla_\lambda\mathcal{Z}\,{}^*R^\lambda{}_{\alpha\beta\rho} + \nabla_\mu\mathcal{Z}R^*_{\nu\alpha\beta\rho}\Big]s^\alpha s^\beta \hat{p}^\mu\hat{p}^\nu\hat{p}^\rho = \mathcal{O}(\mathcal{S}^3), \tag{58}$$

and

$$\Big[4\nabla_\mu N^{\mathrm{NS}}_{\alpha\nu\beta\rho} + 2\nabla_{[\alpha}\mathcal{M}^{(1)}_{|\mu|\nu]\beta\rho} + \big(g_{\alpha\mu}Y_{\lambda\nu} - g_{\mu\nu}Y_{\lambda\alpha}\big)\xi_\kappa \,{}^*R^{\lambda\kappa}{}_{\beta\rho} + \big(2Y_{\alpha\mu}\xi_\lambda - \big(Y_{\lambda\mu}\xi_\alpha + Y_{\alpha\lambda}\xi_\mu\big)$$

$$+ 3g_{\alpha\mu}\nabla_\lambda\mathcal{Z}\big)\,{}^*R^\lambda{}_{\nu\beta\rho} - 3g_{\mu\nu}\nabla_\lambda\mathcal{Z}\,{}^*R^\lambda{}_{\alpha\beta\rho} + 3\nabla_\mu\mathcal{Z}R^*_{\nu\alpha\beta\rho}\Big]s^\alpha s^\beta \hat{p}^\mu\hat{p}^\nu\hat{p}^\rho = \mathcal{O}(\mathcal{S}^3). \tag{59}$$

In the continuation, we will refer to these two problems are respectively the "black hole problem" ($\kappa = 1$) and the "neutron star problem" ($\kappa \neq 1$). Their resolutions are independent and will be addressed separately. Notice that the overall quasi-conserved quantity is given by

$$Q^{(2)} = Q_{\mathrm{R}} + Q_{\mathrm{BH}} + (\kappa - 1)Q_{\mathrm{NS}}. \tag{60}$$

The contributions $Q_{\mathrm{BH}}$ and $Q_{\mathrm{NS}}$ can be directly computed from the corresponding $N_{\alpha\beta\gamma\delta}$ tensor through Eq. (56).

## 4  Kerr covariant formalism: Generalities

In this Section, we will show that the very structure of Kerr spacetime allows us to reduce the differential constraint equations (40)-(58)-(59) to *purely algebraic* relations. It is then possible to find a unique non-trivial solution to the black hole constraint (58), as will be demonstrated in Section 6. It also enables to provide an algebraic way for solving the $\kappa \neq 1$ linear and quadratic problems (*i.e.* Eq. (40) and (59), respectively), as will be discussed in Section 7.

### 4.1  Covariant building blocks for Kerr

In Kerr spacetime, the constraint equations (40)-(58)-(59) can be *fully* expressed in terms of the basic tensors that live on the manifold (that is the metric $g_{\mu\nu}$, the Levi-Civita tensor $\varepsilon_{\mu\nu\rho\sigma}$ and the Kronecker symbol $\delta^\mu_\nu$) and of three additional tensorial structures: the timelike Killing vector field $\xi^\mu$, the complex scalar

$$\mathcal{R} \triangleq r + ia\cos\theta, \tag{61}$$

and the 2-form

$$N_{\alpha\beta} \triangleq -iG_{\alpha\beta\mu\nu}l^\mu n^\nu. \tag{62}$$

We use the convention $\varepsilon_{tr\theta\phi} = -1$ as in [25]. Here, $l^\mu$ and $n^\nu$ are the two principal null directions of Kerr:

$$l^\mu \triangleq \frac{1}{\Delta}\big(r^2 + a^2, \quad \Delta, \quad 0, \quad a\big), \qquad n^\mu \triangleq \frac{1}{2\Sigma}\big(r^2 + a^2, \quad -\Delta, \quad 0, \quad a\big), \tag{63}$$

and $G_{\alpha\beta}{}^{\gamma\delta}$ is (four times) the projector

$$G_{\alpha\beta}{}^{\gamma\delta} \triangleq 2\delta_\alpha^{[\gamma}\delta_\beta^{\delta]} - i\varepsilon_{\alpha\beta}{}^{\gamma\delta}. \tag{64}$$

Notice that we have the property

$$N_{\alpha\beta} = \frac{2}{\xi^2}(\nabla_{[\alpha}\mathcal{R}\xi_{\beta]*} + i\nabla_{[\alpha}\mathcal{R}\xi_{\beta]}). \tag{65}$$

The Killing-Yano and Riemann tensors can be written algebraically in terms of these objects:

$$Y_{\alpha\beta} = -\frac{1}{2}\mathcal{R}N_{\alpha\beta} + c.c., \qquad R_{\alpha\beta\gamma\delta} = M\,\mathrm{Re}\left(\frac{3N_{\alpha\beta}N_{\gamma\delta} - G_{\alpha\beta\gamma\delta}}{\mathcal{R}^3}\right). \tag{66}$$

Moreover, they obey the following closed differential relations,

$$i\nabla_\alpha\mathcal{R} = N_{\alpha\beta}\xi^\beta, \qquad i\nabla_\gamma\left(\mathcal{R}N_{\alpha\beta}\right) = G_{\alpha\beta\gamma\delta}\xi^\delta, \qquad i\nabla_\alpha\xi_\beta = -\frac{M}{2}\left(\frac{N_{\alpha\beta}}{\mathcal{R}^2} - \frac{\bar{N}_{\alpha\beta}}{\bar{\mathcal{R}}^2}\right). \tag{67}$$

All the derivatives appearing in the constraints can consequently be expressed in terms of purely algebraic relations between the covariant building blocks.

## 4.2 Some identities

Let us first derive some useful identities. Many of them can be found in [28,29]. We have the algebraic identities

$$N_{\alpha\beta}N^\beta{}_\gamma = -g_{\alpha\gamma}, \qquad N_{\alpha\beta}\bar{N}^{\alpha\beta} = 0. \tag{68}$$

Notice that this first relation yields

$$N_{\alpha\beta}N^{\alpha\beta} = 4. \tag{69}$$

Both $N_{\alpha\beta}$ and $G_{\alpha\beta}{}^{\gamma\delta}$ are self-dual tensors:

$$N_{\alpha\beta}^* = iN_{\alpha\beta}, \qquad {}^*G_{\alpha\beta}{}^{\gamma\delta} = G^*{}_{\alpha\beta}{}^{\gamma\delta} = iG_{\alpha\beta}{}^{\gamma\delta}. \tag{70}$$

This leads to the relations

$$^*R_{\alpha\beta\gamma\delta} = R^*_{\alpha\beta\gamma\delta} = -M\,\mathrm{Im}\left(\frac{3N_{\alpha\beta}N_{\gamma\delta} - G_{\alpha\beta\gamma\delta}}{\mathcal{R}^3}\right), \qquad \bar{N}^*_{\alpha\beta} = -i\bar{N}_{\alpha\beta}. \tag{71}$$

Given the identities just derived, the only non-trivial contraction of the 2-form that can be written is

$$h_{\mu\nu} \triangleq N_\mu{}^\alpha\bar{N}_{\nu\alpha}. \tag{72}$$

It is a real, symmetric and traceless tensor:

$$h_{\mu\nu} = h_{(\mu\nu)} = \bar{h}_{\mu\nu}, \qquad h^\mu_\mu = 0. \tag{73}$$

Using the previous identities, one shows that

$$\mathcal{Z} = -\frac{1}{2}\mathrm{Im}\left(\mathcal{R}^2\right). \tag{74}$$

This yields

$$\nabla_\alpha \mathcal{Z} = -\operatorname{Re}\left(\mathcal{R}\xi^\lambda N_{\lambda\alpha}\right). \tag{75}$$

The Killing tensor can be written as

$$K_{\mu\nu} = -\frac{1}{2}\left(\operatorname{Re}\left(\mathcal{R}^2\right)g_{\mu\nu} + \left|\mathcal{R}^2\right|h_{\mu\nu}\right). \tag{76}$$

Its trace is simply

$$K = -2\operatorname{Re}\left(\mathcal{R}^2\right). \tag{77}$$

Other useful identities include

$$N_{\lambda\kappa}G^{\lambda\kappa}{}_{\beta\rho} = 4N_{\beta\rho}, \qquad N_{\lambda\kappa}\bar{G}^{\lambda\kappa}{}_{\beta\rho} = \bar{N}_{\lambda\kappa}G^{\lambda\kappa}{}_{\beta\rho} = 0,$$
$$\bar{N}_{\lambda\kappa}\bar{G}^{\lambda\kappa}{}_{\beta\rho} = 4\bar{N}_{\beta\rho}, \qquad h_{\alpha\lambda}N^\lambda{}_\beta = \bar{N}_{\alpha\beta}. \tag{78}$$

## 4.3 Basis of contractions

Our goal is now to rewrite Eqs. (40)-(58)-(59) as *scalar* (that is, fully-contracted) equations involving only contractions between the Kerr covariant building blocks and the dynamical variables $s^\alpha$ and $\hat{p}^\alpha$. We define

$$\mathcal{S}^2 \triangleq s_\alpha s^\alpha, \qquad \mathcal{P}^2 \triangleq -\hat{p}_\alpha \hat{p}^\alpha, \qquad \mathcal{A} \triangleq s_\alpha \hat{p}^\alpha. \tag{79}$$

We will naturally set $\mathcal{P}^2 = 1$ at the end of the computation, but we find useful to keep this quantity explicit in the intermediate algebra. Notice that once the Tulczyjew-Dixon spin supplementary condition has been enforced, the quantity $\mathcal{A}$ will only depend upon the arbitrary part of the relaxed spin vector $s^\alpha$, which is colinear to the linear momentum $\hat{p}^\mu$. Since this contribution to the spin vector is unphysical, the quantity $\mathcal{A}$ is expected to disappear from any physical expression evaluated under TD SSC, but can nevertheless appear in intermediate computations.

We further define the following quantities at least linear in either $\hat{p}^\mu$ or $s^\mu$,

$$A \triangleq N_{\lambda\mu}\xi^\lambda \hat{p}^\mu, \quad B \triangleq N_{\alpha\mu}s^\alpha \hat{p}^\mu, \quad C \triangleq N_{\lambda\alpha}\xi^\lambda s^\alpha, \quad D \triangleq h_{\lambda\alpha}\xi^\lambda s^\alpha, \quad E \triangleq -\xi_\alpha \hat{p}^\alpha, \tag{80a}$$
$$E_s \triangleq -\xi_\alpha s^\alpha, \quad F \triangleq h_{\lambda\mu}\xi^\lambda \hat{p}^\mu, \quad G \triangleq h_{\alpha\mu}s^\alpha \hat{p}^\mu, \quad H \triangleq h_{\mu\nu}\hat{p}^\mu \hat{p}^\nu, \quad I \triangleq h_{\alpha\beta}s^\alpha s^\beta. \tag{80b}$$

Because of the algebraic identities derived above, these scalars form a spanning set of scalars built from contractions among the Kerr covariant building blocks. Any higher order contraction between building blocks will reduce to a product of the ones provided in the above list with coefficients that may depend upon $M$, $a$ and $\mathcal{R}$. The quantities $A, B, C$ are complex while the others are real. Notice that we don't have to include $\xi^2$ in our basis of building blocks, since it is a function of $\mathcal{R}$ and $M$,

$$\xi^2 = -1 + 2M\operatorname{Re}\left(\mathcal{R}^{-1}\right). \tag{81}$$

It can consequently be written in terms of the other quantities.

We further define the following quantity independent from $M$:

$$J \triangleq \xi^\alpha(h_{\alpha\beta} + g_{\alpha\beta})\xi^\beta = \frac{2a^2\sin^2\theta}{r^2 + a^2\cos^2\theta}, \tag{82}$$

where the last expression is evaluated in Boyer-Lindquist coordinates. We can now use $J$ as a Kerr covariant substitute for $a$: we will consider in what follows quantities built from (80), the complex scalar $\mathcal{R}$, the mass $M$ and $J$.

Table 1: Spanning set of elements with $(s,p)^{\pm}$ grading with $s+p=1$ and $s+p=2$.

| $(s,p)^{\pm}$ | Spanning set | Real dimension of the spanning set |
|---|---|---|
| $(1,0)^{+}$ | $C$ | 2 |
| $(1,0)^{-}$ | $D$, $E_s$ | 2 |
| $(0,1)^{+}$ | $A$ | 2 |
| $(0,1)^{-}$ | $E$, $F$ | 2 |
| $(2,0)^{+}$ | $I$, $\mathcal{S}$, $(1,0)^{+}\times(1,0)^{+}$, $(1,0)^{-}\times(1,0)^{-}$ | 8 |
| $(2,0)^{-}$ | $(1,0)^{+}\times(1,0)^{-}$ | 4 |
| $(1,1)^{+}$ | $G$, $\mathcal{A}$, $(1,0)^{+}\times(0,1)^{+}$, $(1,0)^{-}\times(0,1)^{-}$ | 10 |
| $(1,1)^{-}$ | $B$, $(1,0)^{+}\times(0,1)^{-}$, $(1,0)^{-}\times(0,1)^{+}$ | 10 |
| $(0,2)^{+}$ | $H$, $\mathcal{P}$, $(0,1)^{+}\times(0,1)^{+}$, $(0,1)^{-}\times(0,1)^{-}$ | 8 |
| $(0,2)^{-}$ | $(0,1)^{+}\times(0,1)^{-}$ | 4 |

## 4.4 A $\mathbb{Z}_2$ grading

We now define a $\mathbb{Z}_2$ grading $\{,\}$ as follows. We note that the determining equations for the covariant building blocks for Kerr from Eq. (61) to Eq. (78) are invariant under the following $\mathbb{Z}_2$ grading: $\{g_{\alpha\beta}\}=\{M\}=\{x^{\mu}\}=\{\nabla_{\mu}\}=\{\mathcal{R}\}=\{\mathcal{Z}\}=\{G_{\alpha\beta\gamma\delta}\}=\{h_{\mu\nu}\}=\{K_{\mu\nu}\}=+1$ and $\{N_{\alpha\beta}\}=\{\xi^{\alpha}\}=\{Y_{\alpha\beta}\}=-1$.

Further assigning $\{s^{\mu}\}=\{p^{\mu}\}=+1$, we deduce that

$$\{A\}=\{C\}=\{G\}=\{H\}=\{I\}=+1, \qquad \{B\}=\{D\}=\{E\}=\{E_s\}=\{F\}=-1. \tag{83}$$

Since the constraints (58), (59) have grading $+1$, the odd quantities will have to be combined in pairs in order to build a solution to the constraint.

We define the $(s,p)^{\pm}$ grading of an expression as the $s$ numbers of $s^{\alpha}$ and $p$ number of $\hat{p}^{\alpha}$ factors in the expression with the sign $\pm$ indicating the $\mathbb{Z}_2$ grading. The complete list of the lowest $s+p=1$ and $s+p=2$ grading spanning elements is given in Table 1. The list of spanning elements of grading $(s,p)^{\pm}$ for $s+p\geq 3$ is obtained iteratively by direct product of the lower order basis elements. For example, the independent real terms of grading $(2,1)^{+}$ are obtained from $(2,0)^{+}\times(0,1)^{+}$, $(2,0)^{-}\times(0,1)^{-}$, $(1,1)^{+}\times(1,0)^{+}$ and $(1,1)^{-}\times(1,0)^{-}$ with duplicated elements suppressed.

Of prime importance for solving the linear and quadratic constraint equations will be the elements of gradings $(2,2)^{+}$ and $(2,3)^{+}$. Their respective spanning sets contain 118 and 284 elements, which are explicitly listed in the appended *Mathematica* notebooks.

## 4.5 The $\alpha$-$\omega$ basis

We note that the covariant building blocks all depend on $\mathcal{R}$ through real and imaginary parts of expressions containing fractions of $\mathcal{R}$ and $\bar{\mathcal{R}}$. We find therefore natural to define the objects ($n,p\in\mathbb{Z}$ and $K=1,A,B,C,\dots,J$ or any combination of these objects):

$$\alpha_K^{(n,p)}\triangleq\mathrm{Re}\left(\frac{K\bar{\mathcal{R}}^n}{\mathcal{R}^p}\right), \qquad \omega_K^{(n,p)}\triangleq\mathrm{Im}\left(\frac{K\bar{\mathcal{R}}^n}{\mathcal{R}^p}\right). \tag{84}$$

They satisfy the following properties:

$$\alpha_{iK}^{(n,p)}=-\omega_K^{(n,p)}, \qquad \omega_{iK}^{(n,p)}=\alpha_K^{(n,p)}, \qquad \alpha_{\bar{K}}^{(n,p)}=\alpha_K^{(-p,-n)}, \qquad \omega_{\bar{K}}^{(n,p)}=-\omega_K^{(-p,-n)}. \tag{85}$$

Moreover, one has

$$|\mathcal{R}|^2\alpha_K^{(n,p)}=\alpha_K^{(n+1,p-1)}, \qquad \mathrm{Re}\left(\mathcal{R}^2\right)\alpha_K^{(n,p)}=\frac{1}{2}\left[\alpha_K^{(n,p-2)}+\alpha_K^{(n+2,p)}\right],$$

$$\alpha_{\mathcal{R}^kK}^{(n,p)}=\alpha_K^{(n,p-k)}, \qquad \alpha_{\bar{\mathcal{R}}^kK}^{(n,p)}=\alpha_K^{(n+k,p)}. \tag{86}$$

The same properties hold with $\omega$ instead of $\alpha$. Finally, $\alpha$ and $\omega$ are linear in their subscript argument with respect to real-valued functions. Let us denote $\ell^\mu = \hat{p}^\mu$ or $s^\mu$. Then, for any $T \triangleq T_{\mu_1\ldots\mu_k}\ell^{\mu_1}\ldots\ell^{\mu_k}$, we define the operator $\hat{\nabla}$ as

$$\hat{\nabla}T \triangleq \hat{p}^\lambda \nabla_\lambda\big(T_{\mu_1\ldots\mu_k}\big)\ell^{\mu_1}\ldots\ell^{\mu_k}\,. \tag{87}$$

Making use of the identities

$$\hat{\nabla}\mathcal{R}^n = in\mathcal{R}^{n-1}A, \qquad \hat{\nabla}\bar{\mathcal{R}} = -in\bar{\mathcal{R}}^{n-1}\bar{A}, \tag{88}$$

we get the following relations:

$$\hat{\nabla}\alpha_K^{(n,p)} = \alpha_{\hat{\nabla}K}^{(n,p)} + n\omega_{K\bar{A}}^{(n-1,p)} + p\omega_{KA}^{(n,p+1)}, \qquad \hat{\nabla}\omega_K^{(n,p)} = \omega_{\hat{\nabla}K}^{(n,p)} - n\alpha_{K\bar{A}}^{(n-1,p)} - p\alpha_{KA}^{(n,p+1)}\,. \tag{89}$$

We use dimensions such that $G = c = 1$. Given the large amount of definitions, we find useful to summarize the mass dimensions $[\,,]$ of all quantities in order to keep track of the powers of the mass $M$ that can arise. We have the following mass dimensions $[\nabla_\mu] = -1$, $[g_{\mu\nu}] = [\xi^\alpha] = [N_{\alpha\beta}] = [G_{\alpha\beta\mu\nu}] = 0$, $[x^\mu] = [M] = [\mathcal{R}] = [Y_{\alpha\beta}] = 1$ and $[K_{\alpha\beta}] = [K] = 2$. We deduce

$$[X] = 0, \qquad [\alpha_X^{(n,p)}] = [\omega_X^{(n,p)}] = n - p\,, \tag{90}$$

where $X$ is any function of the set $A, B, C, D, E, E_s, F, G, H, I$ defined in Eqs. (80).

## 5 Kerr covariant formalism: reduction of the constraints

### 5.1 Linear constraint

In this section, we will reduce the linear constraint equations in the case where $Y_{\mu\nu}$ is Kerr Killing-Yano tensor. Using the explicit form of $\mathcal{L}_{\mu\nu}$ as defined in (15b) and expressing the quantity $\mathcal{L}_{\mu\nu}^* Y^{\mu\nu}$ in terms of covariant building blocks we find after evaluation

$$\mathcal{L}_{\mu\nu}^* Y^{\mu\nu} = 0, \tag{91}$$

and therefore the $[2,2]$ constraint is automatically fulfilled. The $[2,4]$ constraint can be rewritten

$$-\mu Y_{\alpha\beta}p^{[\alpha}D^{\beta]*\lambda}\hat{p}_\lambda + Y_{\alpha\beta}p^{[\alpha}\mathcal{L}^{\beta]*\lambda}\hat{p}_\lambda = \mathcal{O}(\mathcal{S}^3)\,. \tag{92}$$

A direct computation shows that

$$\mu Y_{\alpha\beta}p^{[\alpha}D^{\beta]*\lambda}\hat{p}_\lambda = -\frac{3M}{2}\big(\mathcal{A}H + \mathcal{P}^2 G\big)\omega_B^{(1,3)}\,, \tag{93}$$

$$Y_{\alpha\beta}p^{[\alpha}\mathcal{L}^{\beta]*\lambda}\hat{p}_\lambda = -\frac{3\kappa M}{2}\big(\mathcal{A}H + \mathcal{P}^2 G\big)\omega_B^{(1,3)}\,. \tag{94}$$

Using these identities and defining $\kappa \triangleq 1 + \delta\kappa$, the $[2,4]$ constraint takes the very simple form

$$-\frac{3M}{2}\delta\kappa\big(\mathcal{A}H + \mathcal{P}^2 G\big)\omega_B^{(1,3)} = \mathcal{O}(\mathcal{S}^3)\,. \tag{95}$$

It is automatically fulfilled for the test body being a black hole, because $\delta\kappa = 0$ in this case. However, if the test body is a neutron star, $\delta\kappa \neq 0$ and the $[2,4]$ constraint is not obeyed anymore. Therefore, $Q_Y$ is not anymore a constant of the motion at second order in the spin in the NS case.

A way to enable the $[2,4]$ constraint to be solvable in the neutron star case is to supplement the Ansatz for the conserved quantity with a term

$$Q_{\text{NS}}^{(1)} = \delta\kappa M_{\alpha\beta\mu\gamma\delta}S^{\alpha\beta}S^{\gamma\delta}p^{\mu}.\tag{96}$$

The conservation equation will then acquire a correction given by

$$\dot{Q}_{\text{NS}}^{(1)} = \delta\kappa v^{\lambda}\nabla_{\lambda}\left(M_{\alpha\beta\mu\gamma\delta}S^{\alpha\beta}S^{\gamma\delta}p^{\mu}\right)\tag{97}$$

$$= \delta\kappa\hat{p}^{\lambda}\nabla_{\lambda}M_{\alpha\beta\mu\gamma\delta}S^{\alpha\beta}S^{\gamma\delta}p^{\mu} + \mathcal{O}(\mathcal{S}^{3})\tag{98}$$

$$= 4\delta\kappa\hat{p}^{\lambda}\nabla_{\lambda}N_{\alpha\beta\mu\gamma\delta}s^{\alpha}\hat{p}^{\beta}s^{\gamma}\hat{p}^{\delta}p^{\mu} + \mathcal{O}(\mathcal{S}^{3}),\tag{99}$$

where $N_{\alpha\beta\mu\gamma\delta} = {}^{\star}M_{\alpha\beta\mu\gamma\delta}^{\star}$. In our scalar notation, it corresponds to supplement the $[2,4]$ constraint with a term $4\delta\kappa\hat{\nabla}N$, with $N$ being of grading $[2,3]$. The constraint to be solved then takes the simple form

$$\boxed{\hat{\nabla}N = \frac{3M}{8}\left(\mathcal{A}H + \mathcal{P}^{2}G\right)\omega_{B}^{(1,3)}.}\tag{100}$$

It is useful to summarize the discussion by the two following statements:

**Main result 1.** *Rüdiger's linear invariant $Q_{Y} = Y_{\alpha\beta}^{*}S^{\alpha\beta}$ is still conserved for the MPD equations at second order in the spin magnitude for spin-induced quadrupoles, i.e. $\dot{Q}_{Y} = \mathcal{O}(\mathcal{S}^{3})$ provided that $\delta\kappa = 0$, i.e. if the test body possesses the multipole structure of a black hole.*

**Preliminary result 1.** *Any tensor $N_{\alpha\beta\mu\gamma\delta}$ possessing the algebraic symmetries*

$$N_{\alpha\beta\mu\gamma\delta} = N_{\gamma\delta\mu\alpha\beta} = N_{[\alpha\beta]\mu\gamma\delta} = N_{\alpha\beta\mu[\gamma\delta]},$$

*and satisfying the constraint equation*

$$\hat{\nabla}N = \frac{3M}{8}\left(\mathcal{A}H + \mathcal{P}^{2}G\right)\omega_{B}^{(1,3)},\tag{101}$$

*will give rise to a quantity*

$$\mathcal{Q}^{(1)} = Q_{Y} + \delta\kappa M_{\alpha\beta\mu\gamma\delta}S^{\alpha\beta}S^{\gamma\delta}p^{\mu}, \qquad M_{\alpha\beta\mu\gamma\delta} = {}^{*}N_{\alpha\beta\mu\gamma\delta}^{*},\tag{102}$$

*which is conserved up to second order in the spin parameter for the spin-induced quadrupole MPD equations* (9), *i.e. $\dot{Q}^{(1)} = \mathcal{O}(\mathcal{S}^{3})$, regardless to the value taken by $\delta\kappa$.*

## 5.2 Quadratic constraint

### 5.2.1 Some identities

Before going further on, it is useful to notice that all the covariant building blocks combinations that will appear in our equations will not be linearly independent. Actually, a direct computation shows that

$$2|B|^{2} + 2\mathcal{A}G + \mathcal{P}^{2}I - \mathcal{S}^{2}H = 0,\tag{103a}$$

$$\left(\mathcal{A}H + \mathcal{P}^{2}G\right)\omega_{C}^{(1,3)} = \left(\mathcal{A}G + \mathcal{P}^{2}I\right)\omega_{A}^{(1,3)} - \left(\mathcal{A}F + \mathcal{P}^{2}D\right)\omega_{B}^{(1,3)}$$
$$+ \left(\mathcal{A}^{2} + \mathcal{P}^{2}\mathcal{S}^{2}\right)\omega_{\tilde{A}}^{(1,3)} + \left(\mathcal{A}E + \mathcal{P}^{2}E_{s}\right)\omega_{\tilde{B}}^{(1,3)},\tag{103b}$$

$$\omega_{\tilde{A}B^{2}}^{(1,3)} = -|B|^{2}\omega_{A}^{(1,3)} - \left(\mathcal{A}F - EG + E_{s}H + \mathcal{P}^{2}D\right)\omega_{B}^{(1,3)}.\tag{103c}$$

Moreover, let us mention that the identities

$$\omega_K^{(0,k)}\alpha_L^{(n,p)} = \frac{1}{2}\Big[\omega_{KL}^{(n,p+k)} - \omega_{\bar{K}L}^{(n-k,p)}\Big], \tag{104}$$

$$\mathrm{Re}\big(\mathcal{R}^2\big)\mathrm{Im}\bigg(\frac{K}{\mathcal{R}^4}\bigg) = \frac{1}{2}\big(\omega_K^{(0,2)} + \omega_K^{(2,4)}\big),$$

$$|\mathcal{R}|^2\mathrm{Im}\bigg(\frac{K}{\mathcal{R}^4}\bigg) = \omega_K^{(1,3)}, \tag{105}$$

will be useful in the following computations.

### 5.2.2 Reducing the $\mathcal{M}^{(1)}$ contribution

Our goal is here to compute the contribution

$$\mathrm{DM} \triangleq 2\nabla_{[\alpha|}\mathcal{M}^{(1)}_{\mu|\nu]\beta\rho}s^{\alpha}s^{\beta}\hat{p}^{\mu}\hat{p}^{\nu}\hat{p}^{\rho}, \tag{106}$$

in some details, as a proof of principle of the computations to follow, which will not be developed in full details. Noticing the identity

$$\nabla_{\mu}K_{\alpha\beta} = 2\varepsilon_{\lambda\rho\mu(\alpha}Y^{\lambda}{}_{\beta)}\xi^{\rho}, \tag{107}$$

we get

$$\nabla_{\mu}\mathcal{M}^{(1)}_{\alpha\nu\beta\rho} = K_{\alpha\lambda}\nabla_{\mu}R^{\lambda}{}_{\nu\beta\rho} + \nabla_{\nu}\mathcal{Z}R^{*}_{\mu\alpha\beta\rho} - \big(Y_{\mu\nu}\xi_{\lambda} + g_{\mu\nu}\nabla_{\lambda}\mathcal{Z} + Y_{\lambda\mu}\xi_{\nu}\big)R^{*\lambda}{}_{\alpha\beta\rho}$$
$$+ \big(2\xi_{\nu}Y_{\lambda\alpha} - 2\xi_{\lambda}Y_{\nu\alpha} + g_{\alpha\nu}\nabla_{\lambda}\mathcal{Z}\big)R^{*\lambda}{}_{\mu\beta\rho} + \big(2g_{\mu\nu}Y_{\kappa\alpha} - g_{\alpha\nu}Y_{\kappa\mu}\big)\xi_{\lambda}R^{*\lambda\kappa}{}_{\beta\rho}. \tag{108}$$

Making use of Eq. (108), one can show that

$$\mathrm{DM} = \bigg[2K_{\mu\lambda}\nabla_{[\alpha}R^{\lambda}{}_{\nu]\beta\rho} + \nabla_{\nu}\mathcal{Z}R^{*}_{\alpha\mu\beta\rho} + \big(2\xi_{\nu}Y_{\lambda\mu} + g_{\mu\nu}\nabla_{\lambda}\mathcal{Z}\big)R^{*\lambda}{}_{\alpha\beta\rho} \tag{109}$$
$$- \big(Y_{\lambda\alpha}\xi_{\mu} + \xi_{\alpha}Y_{\lambda\mu} + g_{\mu\alpha}\nabla_{\lambda}\mathcal{Z}\big)R^{*\lambda}{}_{\nu\beta\rho} + \big(g_{\mu\alpha}Y_{\kappa\nu} - g_{\mu\nu}Y_{\kappa\alpha}\big)\xi_{\lambda}R^{*\lambda\kappa}{}_{\beta\rho}\bigg]s^{\alpha}s^{\beta}\hat{p}^{\mu}\hat{p}^{\nu}\hat{p}^{\rho}.$$

Using the various identities derived above, the relations of Appendix B and performing some simple algebra, one can express this contribution in terms of linearly independent quantities as

$$\mathrm{DM} = -\frac{M}{4}\big(\mathcal{A}^2 + \mathcal{P}^2\mathcal{S}^2\big)\big(5\omega_A^{(0,2)} + 4\omega_{\bar{A}}^{(1,3)} + 3\omega_A^{(2,4)}\big) - \frac{M}{2}\big(\mathcal{A}E + \mathcal{P}^2 E_s\big)\big(\omega_B^{(0,2)} - \omega_{\bar{B}}^{(1,3)} - 3\omega_B^{(2,4)}\big)$$
$$+ \frac{3M}{2}\big(2\mathcal{A}F - EG + E_s H + 2\mathcal{P}^2 D\big)\omega_B^{(1,3)} - \frac{9M}{4}\omega_{AB^2}^{(0,2)} - \frac{15M}{4}\omega_{AB^2}^{(2,4)}. \tag{110}$$

### 5.2.3 The black hole constraint equation

Making use of the notations introduced above, the constraint equation (58) can be written as

$$4\hat{\nabla}N^{\mathrm{BH}} + \mathrm{DM} + \Upsilon = \mathcal{O}\big(\mathcal{S}^3\big), \tag{111}$$

where

$$\Upsilon \triangleq \bigg[\big(g_{\alpha\mu}Y_{\lambda\nu} - g_{\mu\nu}Y_{\lambda\alpha}\big)\xi_{\kappa}{}^{*}R^{\lambda\kappa}{}_{\beta\rho} + \big(2Y_{\alpha\mu}\xi_{\lambda} + \big(Y_{\lambda\mu}\xi_{\alpha} + Y_{\alpha\lambda}\xi_{\mu}\big) + 3g_{\alpha\mu}\nabla_{\lambda}\mathcal{Z}\big)^{*}R^{\lambda}{}_{\nu\beta\rho}$$
$$- 3g_{\mu\nu}\nabla_{\lambda}\mathcal{Z}^{*}R^{\lambda}{}_{\alpha\beta\rho} + \nabla_{\mu}\mathcal{Z}R^{*}_{\nu\alpha\beta\rho}\bigg]s^{\alpha}s^{\beta}\hat{p}^{\mu}\hat{p}^{\nu}\hat{p}^{\rho}. \tag{112}$$

Using the scalar basis introduced above and the identities (103) yields

$$\Upsilon = \frac{M}{2}\left(\mathcal{A}^2 + \mathcal{P}^2\mathcal{S}^2\right)\left[3\omega_A^{(0,2)} + \omega_{\bar{A}}^{(1,3)}\right] + \frac{M}{2}\left(\mathcal{A}E + \mathcal{P}^2 E_s\right)\left[8\omega_B^{(0,2)} - 3\omega_{\bar{B}}^{(1,3)}\right]$$
$$+ \frac{3M}{2}\omega_{AB^2}^{(0,2)} + \frac{9M}{2}|B|^2\omega_A^{(1,3)} - \frac{3M}{2}\left(\mathcal{A}F + \mathcal{P}^2 D\right)\omega_B^{(1,3)}. \tag{113}$$

In summary, Eq. (111) can be written as

$$\hat{\nabla}N^{\mathrm{BH}} = \Upsilon_{\mathrm{BH}}, \tag{114}$$

with the source term

$$\Upsilon_{\mathrm{BH}} = -\frac{\mathrm{D}M + \Upsilon}{4}. \tag{115}$$

### 5.3 The neutron star constraint equation

Repeating the very same procedure, the neutron star constraint (59) reduces to the scalar-like equation

$$\hat{\nabla}N^{\mathrm{NS}} = \Upsilon_{\mathrm{NS}}, \tag{116}$$

with source

$$\Upsilon_{\mathrm{NS}} = \frac{3M}{16}\left(\mathcal{A}^2 + \mathcal{P}^2\mathcal{S}^2\right)\left(\omega_A^{(0,2)} + \omega_A^{(2,4)} + 2\omega_{\bar{A}}^{(1,3)}\right) - \frac{3M}{8}\left(\mathcal{A}E + \mathcal{P}^2 E_s\right)\left(\omega_B^{(0,2)} + \omega_B^{(2,4)}\right)$$
$$+ \frac{15M}{16}\left(\omega_{AB^2}^{(0,2)} + \omega_{AB^2}^{(2,4)}\right) + \frac{15M}{8}\omega_{\bar{A}B^2}^{(1,3)} + \frac{3}{4}M\left(\mathcal{A}F + \mathcal{P}^2 D\right)\omega_B^{(1,3)}. \tag{117}$$

## 6 Solution for the quadratic invariant in the black hole case

We will now try to find a quadratic conserved quantity for the $\delta\kappa = 0$ case. This corresponds to find a solution to the black hole constraint equation (114). In order to reach this goal, we will postulate an Ansatz for the fully-contracted quantity $N$ appearing in the left-hand side of Eq. (114) and then use the covariant building blocks formulation to constrain the Ansatz coefficients.

### 6.1 The Ansatz

Let us consider the following Ansatz

$$N_{\alpha\beta\gamma\delta}^{\mathrm{BH}} \triangleq \sum_{A=1}^{4} \Lambda_A N_{\alpha\beta\gamma\delta}^{(A)}, \tag{118}$$

where $\Lambda_A$ are arbitrary coefficients and where

$$N_{\alpha\beta\gamma\delta}^{(A)} \triangleq {}^*\mathcal{M}_{\alpha\beta\gamma\delta}^{*(A)}. \tag{119}$$

The quantity $\mathcal{M}_{\alpha\beta\gamma\delta}^{(1)}$ has been defined in Eq. (55), and we introduce

$$\mathcal{M}_{\alpha\beta\gamma\delta}^{(2)} \triangleq Y^\lambda{}_\alpha Y^\sigma{}_\gamma R_{\lambda\beta\sigma\delta}, \qquad \mathcal{M}_{\alpha\beta\gamma\delta}^{(3)} \triangleq g_{\alpha\gamma}\xi_\beta\xi_\delta, \qquad \mathcal{M}_{\alpha\beta\gamma\delta}^{(4)} \triangleq g_{\alpha\gamma}g_{\beta\delta}\xi^2. \tag{120}$$

Using the identities derived in Appendix B, one can show that the directional derivatives of the $N^{(A)}$ are given by

$$\hat{\nabla}N^{(1)} = \frac{M}{4}\big(\mathcal{A}^2 + \mathcal{P}^2\mathcal{S}^2\big)\Big(\omega_A^{(0,2)} + 2\omega_{\bar{A}}^{(1,3)} + 3\omega_A^{(2,4)}\Big)$$
$$- \frac{M}{2}\big(\mathcal{A}E + \mathcal{P}^2 E_s\big)\Big(5\omega_B^{(0,2)} - 2\omega_{\bar{B}}^{(1,3)} + 3\omega_B^{(2,4)}\Big) - \frac{9M}{2}|B|^2\omega_A^{(1,3)}$$
$$- \frac{3M}{2}\big(\mathcal{A}F - EG + E_s H + \mathcal{P}^2 D\big)\omega_B^{(1,3)} + \frac{9M}{4}\omega_{AB^2}^{(0,2)} + \frac{15M}{4}\omega_{AB^2}^{(2,4)}, \tag{121a}$$

$$\hat{\nabla}N^{(2)} = -\frac{M}{4}\big(\mathcal{A}^2 + \mathcal{P}^2\mathcal{S}^2\big)\omega_A^{(0,2)} + \frac{M}{2}\big(\mathcal{A}E + \mathcal{P}^2 E_s\big)\omega_B^{(0,2)} - \frac{3M}{4}\omega_{AB^2}^{(0,2)}, \tag{121b}$$

$$\hat{\nabla}N^{(3)} = \frac{M}{2}\Big[\big(\mathcal{S}^2\mathcal{P}^2 + \mathcal{A}^2\big)\omega_A^{(0,2)} + \big(\mathcal{A}E + \mathcal{P}^2 E_s\big)\omega_B^{(0,2)}\Big], \tag{121c}$$

$$\hat{\nabla}N^{(4)} = M\big(\mathcal{S}^2\mathcal{P}^2 + \mathcal{A}^2\big)\omega_A^{(0,2)}. \tag{121d}$$

## 6.2 Solution to the constraint

We will now look for a solution to the black hole constraint equation (114) using the Ansatz (118), *i.e.* we are seeking for specific values of the parameters $\Lambda_A$ such that Eq. (111) is fulfilled. More explicitly, one therefore requires

$$\sum_{A=1}^{4} \Lambda_A DN^{(A)} - \Upsilon_{\text{BH}} \overset{!}{=} 0. \tag{122}$$

The left-hand side of this equation takes the form of a first order polynomial, homogeneous in the ten *linearly independent* elements (as it can be shown through a direct computation)

$$\omega_A^{(0,2)}, \quad \omega_B^{(0,2)}, \quad \omega_{AB^2}^{(0,2)}, \quad \omega_A^{(1,3)}, \quad \omega_B^{(1,3)}, \quad \omega_{\bar{A}}^{(1,3)}, \quad \omega_{\bar{B}}^{(1,3)}, \quad \omega_A^{(2,4)}, \quad \omega_B^{(2,4)}, \quad \omega_{AB^2}^{(2,4)}. \tag{123}$$

Because all the combinations of these elements implied in the constraint equation are linearly independent, all the coefficients appearing in front of these expressions should vanish independently.

All the terms do not appear in all the contributions, as depicted in Table 2. In order to fix the values of the Ansatz coefficients, let us proceed along the following sequence:

⬦ $\underline{\omega_A^{(1,3)}}$ **term**: this contribution reads

$$3M\Big(-6\Lambda_1 + \frac{3}{2}\Big)|B|^2\omega_A^{(1,3)}. \tag{124}$$

Table 2: Structure of the distribution of the different types of contractions in the various contribution to the black hole constraint equation.

| | $\omega_A^{(0,2)}$ | $\omega_B^{(0,2)}$ | $\omega_{AB^2}^{(0,2)}$ | $\omega_A^{(1,3)}$ | $\omega_B^{(1,3)}$ | $\omega_{\bar{A}}^{(1,3)}$ | $\omega_{\bar{B}}^{(1,3)}$ | $\omega_A^{(2,4)}$ | $\omega_B^{(2,4)}$ | $\omega_{AB^2}^{(2,4)}$ |
|---|---|---|---|---|---|---|---|---|---|---|
| $DN^{(1)}$ | ✓ | ✓ | ✓ | ✓ | ✓ | ✓ | ✓ | ✓ | ✓ | ✓ |
| $DN^{(2)}$ | ✓ | ✓ | ✓ | | | | | | | |
| $DN^{(3)}$ | ✓ | ✓ | | | | | | | | |
| $DN^{(4)}$ | ✓ | | | | | | | | | |
| $\Upsilon_{\text{BH}}$ | ✓ | ✓ | ✓ | ✓ | ✓ | ✓ | ✓ | ✓ | ✓ | ✓ |

One therefore requires

$$\Lambda_1 = \frac{1}{4}. \tag{125}$$

$\diamond$ $\omega_{\bar{A}}^{(1,3)}$, $\omega_{\bar{B}}^{(1,3)}$, $\omega_{B}^{(1,3)}$, $\omega_{A}^{(2,4)}$, $\omega_{B}^{(2,4)}$ and $\omega_{AB^2}^{(2,4)}$ **terms**: their coefficients consistently vanish when (125) is fulfilled.

$\diamond$ $\omega_{AB^2}^{(0,2)}$ **term**: this contribution reads

$$3M\left(3\Lambda_1 - \Lambda_2 \frac{1}{4}\right)\omega_{AB^2}^{(0,2)}. \tag{126}$$

Using (125), this yields

$$\Lambda_2 = \frac{1}{2}. \tag{127}$$

$\diamond$ $\omega_{\bar{A}}^{(1,3)}$, $\omega_{\bar{B}}^{(1,3)}$ and $\omega_{B}^{(1,3)}$ **terms:** their coefficients consistently vanish when Eqs. (125) and (127) are fulfilled.

$\diamond$ $\omega_{B}^{(0,2)}$ **term**: this contribution reads

$$M\left(\mathcal{A}E + \mathcal{P}^2 E_s\right)\left(-10\Lambda_1 + 2\Lambda_2 + 2\Lambda_3 + \frac{7}{2}\right)\omega_{B}^{(0,2)}. \tag{128}$$

Using (125) and (127), this yields

$$\Lambda_3 = -1. \tag{129}$$

$\diamond$ $\omega_{A}^{(0,2)}$ **term**: this contribution reads

$$M\left(\mathcal{A}^2 + \mathcal{P}^2 \mathcal{S}^2\right)\left(\Lambda_1 - \Lambda_2 + 2\Lambda_3 + 4\Lambda_4 + \frac{1}{4}\right)\omega_{A}^{(0,2)}. \tag{130}$$

Using Eqs. (125), (127) and (129), this finally yields

$$\Lambda_4 = \frac{1}{2}. \tag{131}$$

In conclusion, the Ansatz (118) gives a coherent solution to the constraint equation (114) only if

$$\Lambda_1 = \frac{1}{4}, \quad \Lambda_2 = \frac{1}{2}, \quad \Lambda_3 = -1, \quad \Lambda_4 = \frac{1}{2}. \tag{132}$$

More explicitly, it corresponds to set

$$M_{\alpha\beta\gamma\delta} = -g_{\alpha\gamma}\left(\xi_\beta \xi_\delta - \frac{1}{2}g_{\beta\delta}\xi^2\right) + \frac{1}{2}Y_\alpha{}^\lambda\left(Y_\gamma{}^\kappa R_{\lambda\beta\kappa\delta} + \frac{1}{2}Y_\lambda{}^\kappa R_{\kappa\beta\gamma\delta}\right). \tag{133}$$

## 6.3   Uniqueness of the solution

We now address the uniqueness to the non-trivial solution (133) to the constraint (114) derived above. If one adds an additional piece to our Ansatz, it will satisfy an homogeneous equation since all source terms have been cancelled by the Ansatz. Demonstrating uniqueness of the non-trivial solution (133) therefore amounts to prove that

$$\nabla_{(\mu} N_{\nu|(\alpha\beta)|\rho)} = 0 \,, \tag{134}$$

does not admit any non-trivial solution in Kerr spacetime. We call such a tensor field a Young tableau $(2,2)$ Killing tensor. A trivial Killing tensor is defined as a Killing tensor which is given by a cross-product. Such a trivial Killing tensor would add to the quadratic conserved quantity a product of conserved quantities that are already defined. There is only trivial Killing tensor of symmetry type $(2,2)$ namely $N_{\mu\nu\alpha\beta} = Y_{\mu\nu} Y_{\alpha\beta}$ which correspond to add the product $(Q_Y)^2$ defined in (48) to the quadratic conserved quantity. We checked explicitly by solving the partial differential equations analytically using a *Mathematica* notebook that no non-trivial such tensor exists in a perturbative series expansion in $a$ around $a = 0$ assuming that it only depends upon $r$ and $\theta$.

## 6.4   Summary of the results

Let us summarize the results we have obtained about the quadratic invariants. Our discussion can be compactified in the two following propositions:

**Main result 2.** *The quadratic invariant*

$$Q_{BH}^{(2)} = Q_R + \left[ -g_{\alpha\gamma} \left( \xi_\beta \xi_\delta - \frac{1}{2} g_{\beta\delta} \xi^2 \right) + \frac{1}{2} Y_\alpha{}^\lambda \left( Y_\gamma{}^\kappa R_{\lambda\beta\kappa\delta} + \frac{1}{2} Y_\lambda{}^\kappa R_{\kappa\beta\gamma\delta} \right) \right] S^{\alpha\beta} S^{\gamma\delta} \,, \tag{135}$$

*is conserved for the MPD equations at second order in the spin magnitude for spin-induced quadrupole, i.e. $\dot{Q}_{BH}^{(2)} = \mathcal{O}(\mathcal{S}^3)$ provided that $\delta\kappa = 0$, i.e. if the test body possesses the multipole structure of a black hole. Here, $Q_R = K_{\mu\nu} p^\mu p^\nu + L_{\mu\nu\rho} S^{\mu\nu} p^\rho$ with $L_{\mu\nu\rho}$ given in Eq. (53) is Rüdiger's quadratic invariant [11, 25].*

**Preliminary result 2.** *Any tensor $N_{\alpha\beta\gamma\delta}$ possessing the same algebraic symmetries than the Riemann tensor and satisfying the constraint equation*

$$\hat{\nabla} N = \Upsilon_{NS} \,, \tag{136}$$

*where the source term $\Upsilon_{NS}$ is given in Eq. (117) will give rise to a quantity*

$$Q^{(2)} = Q_{BH}^{(2)} + \delta\kappa M_{\alpha\beta\gamma\delta} S^{\alpha\beta} S^{\gamma\delta} \,, \qquad M_{\alpha\beta\gamma\delta} = {}^*N^*_{\alpha\beta\gamma\delta} \,, \tag{137}$$

*which is conserved up to second order in the spin parameter for the spin-induced quadrupole MPD equations (9), i.e. $\dot{Q}^{(2)} = \mathcal{O}(\mathcal{S}^3)$, regardless to the value taken by $\delta\kappa$.*

We notice that $\Upsilon_{NS}\big|_{a=0} = 0$ in the Schwarzschild case by explicit evaluation of (117). A direct consequence is that the deformation of Rüdiger's quadratic invariant constructed in the black hole case is still quasi-conserved for arbitrary $\kappa$:

**Main result 3.** *In Schwarschild spacetime ($a = 0$), the deformation of Rüdiger's quadratic invariant $Q_{BH}^{(2)}$ given in Eq. (135) is still conserved for the MPD equations up to $O(\mathcal{S}^3)$ corrections for arbitrary ($\kappa \in \mathbb{R}$) spin-induced quadrupole.*

Notice that the conservation does not hold for Rüdiger's linear invariant $Q_Y$. The Kerr case $a \neq 0$ will be further discussed in Section 7.

Table 3: Source terms for the various constraint equations $\hat{\nabla} N = \Upsilon$ studied in the paper.

| PROBLEM | SOURCE TERM $\Upsilon$ | EQUATION | $\Upsilon\big|_{a=0}$ | SOURCE GRADING |
|---|---|---|---|---|
| NS linear invariant | $\Upsilon_{\text{lin}}$ | (101) | $\neq 0$ | $[2,4]^+$ |
| BH quadratic invariant | $\Upsilon_{\text{BH}}$ | (116) | $\neq 0$ | $[2,3]^+$ |
| NS quadratic invariant | $\Upsilon_{\text{NS}}$ | (117) | $0$ | $[2,3]^+$ |

# 7 Neutron star case around Kerr: A no-go result

We summarize in Table 3 the three constraint equations discussed previously. They all take the form

$$\hat{\nabla} N = \Upsilon, \tag{138}$$

with $\Upsilon$ being of grading $[s,p]^+$. It implies that $N$ should be of grading $[s,p-1]^+$.

Let $\{K_{\mathfrak{a}}\}$ be a basis of linearly independent and dimensionless functions build from the (manifestly real) functions $\mathcal{A}, \mathcal{P}^2, \mathcal{S}^2, \text{Re}\,A, \text{Im}\,A, \ldots, \text{Re}\,C, \text{Im}\,C, D, \ldots, I$. Given that $N$ is dimensionless and given the structure of the source terms $\Upsilon$, we propose the following Ansatz,

$$N = \sum_{\mathfrak{a}} \sum_{(k,l)\in\mathbb{Z}^2} K_{\mathfrak{a}} M^l \Big( C_{\mathfrak{a}}^{(k,l)} f_{\mathfrak{a}}^{(k,l)}(J) \alpha_1^{(k,k+l)} + D_{\mathfrak{a}}^{(k,l)} g_{\mathfrak{a}}^{(k,l)}(J) \omega_1^{(k,k+l)} \Big). \tag{139}$$

Here, $C_{\mathfrak{a}}^{k,l}$ and $D_{\mathfrak{a}}^{k,l}$ are numerical coefficients and $f_{\mathfrak{a}}^{k,l}(J)$ and $g_{\mathfrak{a}}^{k,l}(J)$ are smooth functions of $J$.

We can work with dimensionless quantities by first introducing the dimensionless variables

$$\tilde{r} = \frac{r}{M}, \qquad \tilde{a} = \frac{a}{M}. \tag{140}$$

We notice that the $K_{\mathfrak{a}}$'s are left unchanged and do not depend anymore on $M$, whereas $\mathcal{R} \triangleq M\tilde{\mathcal{R}}$, with $\tilde{\mathcal{R}} \triangleq \tilde{r} + i\tilde{a}$. This yields

$$\alpha_1^{(n,p)} = M^{n-p} \text{Re}\left(\frac{\bar{\tilde{\mathcal{R}}}}{\tilde{\mathcal{R}}}\right) \triangleq M^{n-p} \tilde{\alpha}^{(n,p)}, \qquad \omega_1^{(n,p)} = M^{n-p} \text{Im}\left(\frac{\bar{\tilde{\mathcal{R}}}}{\tilde{\mathcal{R}}}\right) \triangleq M^{n-p} \tilde{\omega}^{(n,p)}. \tag{141}$$

Each derivative of the term present in the Ansatz scales as $M^{-1}$ times a manifestly dimensionless quantity. All the source terms appearing earlier can be written as $\Upsilon = M^{-1}\tilde{\Upsilon}$, with $\tilde{\Upsilon}$ being an dimensionless quantity. This implies that Eq. (139) reduces to

$$N = \sum_{\mathfrak{a}} \sum_{(k,l)\in\mathbb{Z}^2} K_{\mathfrak{a}} \Big( C_{\mathfrak{a}}^{k,l} f_{\mathfrak{a}}^{k,l}(J) \tilde{\alpha}^{(k,k+l)} + D_{\mathfrak{a}}^{k,l} g_{\mathfrak{a}}^{k,l}(J) \tilde{\omega}^{(k,k+l)} \Big), \tag{142}$$

which contain only terms that are explicitly independent of $M$. We can further define $\tilde{\nabla} = M\hat{\nabla}$ the dimensionless derivative operator and the constraints take the dimensionless form $\tilde{\nabla} N = \tilde{\Upsilon}$.

## 7.1 Perturbative expansion in $a$ of the constraint equations

Instead of addressing the non-linear problem in $a$ we will perform a perturbative series in $a$. For any smooth function $f$ of $a$, we define

$$(f)_n \triangleq \frac{d^n f}{da^n}\bigg|_{a=0}. \tag{143}$$

The constraint equation then becomes an infinite hierarchy of equations

$$\left(\tilde{\nabla}N\right)_n = \tilde{\Upsilon}_n, \qquad \forall n \geq 0. \tag{144}$$

Let us describe the $n = 0$ and $n = 1$ equations. Since $J \propto a^2$, the functions $f_{\mathfrak{a}}^{k,l}(J)$ and $g_{\mathfrak{a}}^{k,l}(J)$ do not contribute and can be set to one without loss of generality.

**$n = 0$ equation.** Noticing the identities

$$\tilde{\omega}^{(k,k+l)}\Big|_{a=0} = \tilde{\alpha}_A^{(k,k+l+1)}\Big|_{a=0} = \tilde{\alpha}_{\bar{A}}^{(k-1,k+l)}\Big|_{a=0} = 0, \tag{145}$$

$$\tilde{\alpha}^{(k,k+l)}\Big|_{a=0} = \tilde{r}^{-l}, \qquad \tilde{\omega}_A^{(k,k+l+1)}\Big|_{a=0} = -p_r \tilde{r}^{-(l+1)}, \qquad \tilde{\omega}_{\bar{A}}^{(k-1,k+l)}\Big|_{a=0} = p_r \tilde{r}^{-(l+1)}, \tag{146}$$

and making use of Eq. (89), the $n = 0$ constraint becomes

$$\sum_{\mathfrak{a}} \sum_{(k,l)\in\mathbb{Z}^2} C_{\mathfrak{a}}^{(k,l)}\left[\left(\tilde{\nabla}K_{\mathfrak{a}}\right)_0 \tilde{r}^{-l} - l(K_{\mathfrak{a}})_0 p_r \tilde{r}^{-(l+1)}\right] = \left(\tilde{\Upsilon}\right)_0. \tag{147}$$

It does not depend on the terms involving $\omega$'s contributions. Moreover, denoting

$$C_{\mathfrak{a}}^{(l)} \triangleq \sum_{k\in\mathbb{Z}} C_{\mathfrak{a}}^{(k,l)}, \tag{148}$$

this equation can be further simplified to

$$\sum_{\mathfrak{a}} \sum_{l\in\mathbb{Z}} C_{\mathfrak{a}}^{(l)}\left[\left(\tilde{\nabla}K_{\mathfrak{a}}\right)_0 - l(K_{\mathfrak{a}})_0 p_r \tilde{r}^{-1}\right]\tilde{r}^{-l} = \left(\tilde{\Upsilon}\right)_0. \tag{149}$$

**$n = 1$ equation.** Following an identical procedure and denoting

$$D_{\mathfrak{a}}^{(l)} \triangleq \sum_{k\in\mathbb{Z}} (2k+l)D_{\mathfrak{a}}^{(k,l)}, \tag{150}$$

the $n = 1$ constraint equation can be shown to take the form

$$\sum_{\mathfrak{a}} \sum_{l\in\mathbb{Z}} \Bigg\{ C_{\mathfrak{a}}^{(l)}\left[\left(\tilde{\nabla}K_{\mathfrak{a}}\right)_1 - (K_{\mathfrak{a}})_1 l p_r \tilde{r}^{-1}\right]\tilde{r}^{-l}$$
$$- D_{\mathfrak{a}}^{(l)}\left[\left(\tilde{\nabla}K_{\mathfrak{a}}\right)_0 x + (K_{\mathfrak{a}})_0\left(p_\theta - (l+1)p_r x \tilde{r}^{-1}\right)\right]\tilde{r}^{-(l+1)} \Bigg\} = \left(\tilde{\Upsilon}\right)_1. \tag{151}$$

**Numerical evaluation.** Eqs. (149) and (151) have been numerically evaluated using *Mathematica* in order to try to fix the values of the coefficients $C_{\mathfrak{a}}^{(l)}$ and $D_{\mathfrak{a}}^{(l)}$ that would enable a possible solution to the neutron star cases. We have only looked for "polynomial" solutions to these equations, *i.e.* solutions for which the coefficients of the ansatz are non-vanishing only over a finite interval $[l_{\min}, l_{\max}]$. Given the size of the expressions involved, the only computationally reasonable solving method available to us was the following: let us denote $N$ the number of terms present in the left-hand side of (149) (resp. (151)) for a given $[l_{\min}, l_{\max}]$. Eq. (149) (resp. (151)) was then evaluated $N+1$ times at different random values of its variables and parameters, resulting into a linear system of $N+1$ algebraic equations in $N$ variables (the coefficients $C_{\mathfrak{a}}^{(l)}$ and $D_{\mathfrak{a}}^{(l)}$) that was then solved using the built-in numerical equation solver of *Mathematica*.

This procedure has been proof-tested by reproducing the coefficients corresponding to the black hole quadratic invariant from the source term $\Upsilon_{\mathrm{BH}}$. It has then been used to attempt to find a solution to both the linear and the quadratic neutron star problems, with $[l_{\min}, l_{\max}] = [-10, 100]$. No solution has been found, discarding *a priori* the existence of polynomial-type solutions.

In the *Mathematica* notebooks appended as supplementary material, the interval of $l$ is reduced to $[l_{\min}, l_{\max}] = [0, 5]$ in order to reduce the computational time for the interested reader. The four notebooks related to this section are:

⋄ `Quadratic_BH_LO_final.nb`: check of the numerical evaluation of the $n = 0$ equation: reproduction of the black hole quadratic invariant from $\Upsilon_{\mathrm{BH}}$;

⋄ `Linear_NS_LO_final.nb`: attempt of finding a polynomial solution to the $n = 0$ equation for the neutron star linear problem (source term $\Upsilon_{\mathrm{lin}}$);

⋄ `Quadratic_BH_NLO_final.nb`: check of the numerical evaluation of the $n = 1$ equation: reproduction of the black hole quadratic invariant from $\Upsilon_{\mathrm{BH}}$;

⋄ `Quadratic_NS_NLO_final.nb`: attempt of finding a polynomial solution to the $n = 1$ equation for the neutron star quadratic problem (source term $\Upsilon_{\mathrm{NS}}$).

# 8 Discussion and Outlook

The main results obtained in this paper are as follows. At second order in the spin magnitude and for spin-induced quadrupole with black hole-type coupling ($\kappa = 1$): (i) the linear Rüdiger invariant $Q_Y$ is still quasi-conserved and (ii) a deformation of Rüdiger's quadratic invariant $Q_{\mathrm{BH}}^{(2)}$ exists such that the deformed Rüdiger's quadratic invariant is also quasi-conserved. Finally, (iii) the quasi-conservation of the deformed quadratic invariant can only be extended to arbitrary coupling ($\kappa \neq 1$) around the Schwarszchild spacetime ($a = 0$).

All our attempts to find solutions to the constraint equations in the case of an arbitrary spin-induced coupling in generic Kerr spacetime have failed. Let us notice that, even if someone would succeed in solving them, the quasi-invariants so obtained would not be of direct astrophysical interest. This arises from the fact that, except in the special case where the test body is itself a black hole, the spin-induced term is not the only contribution to the quadrupole. Tidal-type contributions will also arise which will break the quasi-conservation obtained for spin-induced quadrupoles only.

Various extensions of this work would be interesting to explore. Firstly, one could investigate whether deformations of the quasi-invariants studied in this paper still exist at higher orders in the multipolar expansion for black-hole-type couplings. At the next cubic order in the test black hole's spin, which includes its octupole moment, the appropriate equations of motion have been argued to be fixed without ambiguity from appropriate symmetries and after matching to the stationary Kerr solution as at the spin-squared/quadrupolar level, see e.g. [30]. One could then construct ansätze for deformations of the hidden-symmetry constants, and search for solutions that are conserved assuming such equations of motion at this order. At even higher orders, the same considerations alone do not fix the equations of motion, due to the relevance of quadratic-in-curvature couplings at fourth order in spin, but one could still proceed analogously with parametrized equations, perhaps even obtaining constraints on the equations of motion from the existence of conservation laws.

Another possible direction would be to understand the link between the existence of these quasi-conserved quantities and the separability of the associated Hamilton-Jacobi equation at second order in the spin magnitude, thus pushing the analysis of Witzany [31] to the next order

in the multipole expansion. This would also enable to compute the corresponding shifts in the fundamental frequencies of the action-angle variables description of the finite size particle, which are of direct relevance for modeling EMRIs involving spinning secondaries. It would also be enlightening to explore the relationship between the new constants for test black holes in Kerr found in this paper and those for arbitrary-mass-ratio binary black holes at second-post-Newtonian order (including the spin-induced quadrupole effects) found by Tanay *et al.* in Ref. [32], ensuring the integrablity of the system at that order. Finally, if the structure of the 4-dimensional Kerr covariant building blocks can be generalized to higher dimensions and more generic spacetimes, it could bring new insights on hidden symmetries of such spacetimes [33].

## Acknowledgments

A. D. is a Research Fellow and G. C. is Senior Research Associate of the F.R.S.-FNRS. JV is grateful to Lars Andersson, Éanna Flanagan, Abraham Harte and Leo Stein for helpful discussions, and especially to Steffen Aksteiner for confirmation via spinorial methods of the conservation of the generalized Carter constant for a quadrupolar test black hole in Kerr. We thank our referee Scott Hughes for interesting comments and especially for pointing out a mistake in a coefficient.

**Funding information** G. C. acknowledges support from the FNRS research credit No. J003620F and the IISN convention No. 4.4503.15.

## A  Derivation of the quadratic quantity constraint

The aim of this appendix is to provide a derivation of the reduced expression (54) for the constraint of grading $[2,3]$ for the quadratic conserved quantity (31).

Let us denote by $\langle f \rangle_{(n)}$ the terms contained in $f$ that are homogeneous of order $\mathcal{O}(\mathcal{S}^n)$. We now take for granted the validity and uniqueness of the solution (53), as well as the vanishing of the $O(S^0)$ and $O(S^1)$ terms in Eq. (52). The $\mathcal{O}(\mathcal{S}^2)$ constraint equation takes the form

$$\langle \dot{\mathcal{Q}} \rangle_{(2)} = \langle \dot{\mathcal{Q}}_R \rangle_{(2)} + \langle \dot{\mathcal{Q}}^{\text{quad}} \rangle_{(2)} \overset{!}{=} 0 \,. \tag{A.1}$$

We will compute these two contributions separately.

### A.1  Terms coming from $\mathcal{Q}^{\text{quad}}$

Using Eqs. (9b), (18) and (19), the variation of $\mathcal{Q}^{\text{quad}}$ along the trajectory is given by

$$\dot{\mathcal{Q}}^{\text{quad}} = \frac{\mathrm{D}}{\mathrm{d}\tau} M_{\alpha\beta\gamma\delta} S^{\alpha\beta} S^{\gamma\delta} + 2 \frac{\mathrm{D}S^{\alpha\beta}}{\mathrm{d}\tau} M_{\alpha\beta\gamma\delta} S^{\gamma\delta} \tag{A.2a}$$

$$= v^\lambda \left( \nabla_\lambda M_{\alpha\beta\gamma\delta} S^{\alpha\beta} S^{\gamma\delta} + 4 M_{\alpha\beta\gamma\lambda} S^{\alpha\beta} p^\gamma \right) + \mathcal{O}(\mathcal{S}^3) \tag{A.2b}$$

$$= \hat{p}^\lambda \left( \nabla_\lambda M_{\alpha\beta\gamma\delta} S^{\alpha\beta} S^{\gamma\delta} + 4 M_{\alpha\beta\gamma\lambda} S^{\alpha\beta} p^\gamma \right) + \mathcal{O}(\mathcal{S}^3) \tag{A.2c}$$

$$= \nabla_\lambda M_{\alpha\beta\gamma\delta} \hat{p}^\lambda S^{\alpha\beta} S^{\gamma\delta} + \mathcal{O}(\mathcal{S}^3) \,. \tag{A.2d}$$

Recalling that $S^{\alpha\beta} = 2s^{[\alpha}\hat{p}^{\beta]*}$, we obtain the following equation in terms of the independent variables $s_\alpha$ and $p^\mu$:

$$\langle \dot{\mathcal{Q}}^{\text{quad}} \rangle_{(2)} = 4 \nabla_\mu {}^*M^{*\,\alpha\,\,\beta}_{\,\,\,\nu\,\,\rho} s_\alpha s_\beta \hat{p}^\mu \hat{p}^\nu \hat{p}^\rho = 4 \nabla_\mu N_{\alpha\nu\beta\rho} s^\alpha s^\beta \hat{p}^\mu \hat{p}^\nu \hat{p}^\rho \,. \tag{A.3}$$

## A.2 Terms coming from $\mathcal{Q}_R$

One can perform the splitting

$$\left\langle \dot{\mathcal{Q}}_R \right\rangle_{(2)} = \left\langle \dot{\mathcal{Q}}_R^{MD} \right\rangle_{(2)} + \left\langle \dot{\mathcal{Q}}_R^{Q} \right\rangle_{(2)}. \tag{A.4}$$

Here, the "monopole-dipole" terms $\dot{\mathcal{Q}}_R^{MD}$ are those that where already present in [25]:

$$\dot{\mathcal{Q}}_R^{MD} \triangleq \mu^2 D^\lambda{}_\rho \nabla_\lambda K_{\mu\nu} \hat{p}^\mu \hat{p}^\nu \hat{p}^\rho - \frac{\mu}{2} L_{\mu\nu\rho} S^{\mu\nu} R^\rho{}_{\alpha\beta\gamma} \hat{p}^\alpha S^{\beta\gamma} + 2\mu^2 L_{\mu\lambda\rho} D^\lambda{}_\nu \hat{p}^\mu \hat{p}^\nu p^\rho \tag{A.5a}$$

$$= 4\mu {}^*W^{*\alpha}{}_{\mu\nu}{}^{\beta}{}_{\rho} s_\alpha s_\beta \hat{p}^\mu \hat{p}^\nu \hat{p}^\rho + O(\mathcal{S}^3), \tag{A.5b}$$

where (see Eq. (73) of [25])

$$ {}^*W^*{}_{\alpha\beta\gamma\delta\varepsilon} = -\frac{1}{2} {}^*L_{\alpha\beta\lambda} R^{*\lambda}{}_{\gamma\delta\varepsilon}. \tag{A.6}$$

The *relaxed spin vector* $s^\alpha$ is defined from $S^\alpha \triangleq \Pi^\alpha_\beta s^\beta$ where the part of **s** aligned with **p** is left arbitrary, but is assumed (without loss of generality) to be of the same order of magnitude.

The tensor $L_{\alpha\beta\gamma}$ is defined in Eq. (53). In order to simplify Eq. (A.6), we have on the one hand

$$ {}^*\varepsilon_{\alpha\beta\lambda\rho} \nabla^\rho \mathcal{Z} = -2g_{\lambda[\alpha} \nabla_{\beta]} \mathcal{Z}. \tag{A.7}$$

On the other hand,

$$\nabla_{[\alpha} K_{\beta]*\lambda} = 2Y_{\lambda[\alpha} \xi_{\beta]} + 3g_{\lambda[\alpha} Y^\rho{}_{\beta} \xi_{\rho]} = 3Y_{\lambda[\alpha} \xi_{\beta]} + g_{\lambda[\alpha} \nabla_{\beta]} \mathcal{Z}, \tag{A.8}$$

where $\xi^\alpha = -\frac{1}{3} \nabla_\lambda Y^{*\lambda\alpha}$ is the timelike Killing vector associated with the Killing-Yano tensor. Gathering these pieces together, we obtain the reduced expression

$$ {}^*W^*{}_{\alpha\beta\gamma\delta\varepsilon} s^\alpha s^\beta \hat{p}^\mu \hat{p}^\nu \hat{p}^\rho = -\left[ R^{*\lambda}{}_{\gamma\delta\varepsilon} Y_{\lambda[\alpha} \xi_{\beta]} + \nabla_{[\alpha} \mathcal{Z} R^*_{\beta]\gamma\delta\varepsilon} \right] s^\alpha s^\beta \hat{p}^\mu \hat{p}^\nu \hat{p}^\rho. \tag{A.9}$$

The monopole-dipole piece is consequently given by

$$\left\langle \dot{\mathcal{Q}}_R^{MD} \right\rangle_{(2)} = \left( 2(Y_{\lambda\mu} \xi_\alpha - Y_{\lambda\alpha} \xi_\mu) R^{*\lambda}{}_{\nu\beta\rho} - 2\nabla_\mu \mathcal{Z} R^*_{\nu\alpha\beta\rho} \right) s^\alpha s^\beta \hat{p}^\mu \hat{p}^\nu \hat{p}^\rho. \tag{A.10}$$

The "quadrupolar" terms $\dot{\mathcal{Q}}_R^Q$ are the ones induced by the presence of the quadrupole, namely

$$\dot{\mathcal{Q}}_R^Q \triangleq 2\mu K_{\mu\nu} \hat{p}^\mu \mathcal{F}^\nu - \mu \mathcal{L}^\lambda{}_\rho \nabla_\lambda K_{\mu\nu} \hat{p}^\mu \hat{p}^\nu \hat{p}^\rho - 2\mu L_{\mu\lambda\rho} \mathcal{L}^\lambda{}_\nu \hat{p}^\mu \hat{p}^\nu \hat{p}^\rho + \mu L_{\mu\nu\rho} \mathcal{L}^{\mu\nu} \hat{p}^\rho. \tag{A.11}$$

Considering only the spin-induced quadrupole (12) and making use of the identities (15a) to (15c) as well as the explicit form of the tensor $L_{\mu\nu\rho}$ (53), we obtain the reduced expression

$$\left\langle \dot{\mathcal{Q}}_R^Q \right\rangle_{(2)} = \kappa \left[ K_{\mu\lambda} \nabla^\lambda R_{\nu\alpha\beta\rho} - \nabla_\lambda K_{\mu\nu} R^\lambda{}_{\alpha\beta\rho} - \frac{4}{3} \nabla_{[\alpha} K_{\lambda]\nu} R^\lambda{}_{\mu\beta\rho} - \frac{8}{3} \varepsilon_{\alpha\gamma\rho\lambda} \nabla^\lambda \mathcal{Z} R^\gamma{}_{\mu\beta\nu} \right] \Theta^{\alpha\beta} \hat{p}^\mu \hat{p}^\nu \hat{p}^\rho. \tag{A.12}$$

Thanks to the orthogonality condition $p_\alpha S^\alpha = 0$, we can substitute $\Theta^{\alpha\beta}$ in this expression with $\theta^{\alpha\beta}$ defined as

$$\theta^{\alpha\beta} = \Pi^{\alpha\beta} \mathcal{S}^2 - s^\alpha s^\beta. \tag{A.13}$$

After a few algebraic manipulations and making use of Bianchi identities, we obtain

$$\left\langle \dot{\mathcal{Q}}_R^Q \right\rangle_{(2)} = \kappa \left[ \left( \nabla_\nu \left( K_{\mu\lambda} R^\lambda{}_{\alpha\beta\rho} \right) + \nabla_\nu K_{\mu\lambda} R^\lambda{}_{\alpha\beta\rho} - (\nu \leftrightarrow \alpha) \right) \right.$$
$$\left. + \frac{4}{3} \nabla_{[\alpha} K_{\mu]\lambda} R^\lambda{}_{\nu\beta\rho} - \frac{16}{3} \nabla_\lambda \mathcal{Z} g_{\mu[\nu} {}^*R^\lambda{}_{\alpha]\beta\rho} + \frac{8}{3} \nabla_\mu \mathcal{Z} {}^*R_{\nu\alpha\beta\rho} \right] \theta^{\alpha\beta} \hat{p}^\mu \hat{p}^\nu \hat{p}^\rho. \tag{A.14}$$

It is possible to further reduce this expression. It is useful to first derive some properties of the dualizations of Riemann tensor *in Ricci-flat spacetimes*. For any tensor $M_{abcd}$ with the symmetries of the Riemann tensor, one has

$$^*M^{*\alpha\beta}{}_{\mu\nu} = -6\delta^{\alpha\beta ab}_{[\mu\nu cd]}M_{ab}{}^{cd}.$$
(A.15)

In Ricci-flat spacetimes, this yields for the Riemann tensor,

$$^*R^*{}_{\alpha\beta\gamma\delta} = -R_{\alpha\beta\gamma\delta}.$$
(A.16)

Moreover, dualizing this equation one more time gives rise to the identity

$$R^*_{\alpha\beta\gamma\delta} = {}^*R_{\alpha\beta\gamma\delta}.$$
(A.17)

Using $R_{\alpha\beta\gamma\delta} = R_{\gamma\delta\alpha\beta}$, we also deduce

$$R^*_{\alpha\beta\gamma\delta} = R^*_{\gamma\delta\alpha\beta}.$$
(A.18)

Notice that we also have the identity

$$\nabla^\lambda R_{\lambda\alpha\beta\gamma} = 0.$$
(A.19)

Making use of the properties of the Riemann tensor and of the identity $^*R_{\lambda\alpha\beta\rho}g^{\alpha\beta} = 0$, we get the additional relations

$$R_{\lambda\alpha\beta\rho}\theta^{\alpha\beta}\hat{p}^\rho = -R_{\lambda\alpha\beta\rho}s^\alpha s^\beta\hat{p}^\rho,$$
(A.20a)

$$^*R_{\lambda\alpha\beta\rho}\theta^{\alpha\beta}\hat{p}^\rho = -{}^*R_{\lambda\alpha\beta\rho}s^\alpha s^\beta\hat{p}^\rho,$$
(A.20b)

$$R_{\lambda\nu\beta\rho}\theta^{\alpha\beta}\hat{p}^\nu\hat{p}^\rho = R_{\lambda\nu\beta\rho}\big(g^{\alpha\beta}\mathcal{S}^2 - s^\alpha s^\beta\big)\hat{p}^\nu\hat{p}^\rho.$$
(A.20c)

This allows to express $\big\langle\dot{\mathcal{Q}}^Q_R\big\rangle_{(2)}$ in terms of the independent variables as

$$\big\langle\dot{\mathcal{Q}}^Q_R\big\rangle_{(2)} = -\kappa\bigg[\Big(\nabla_\nu\big(K_{\mu\lambda}R^\lambda{}_{\alpha\beta\rho}\big) + \nabla_\nu K_{\mu\lambda}R^\lambda{}_{\alpha\beta\rho} - (\nu\leftrightarrow\alpha)\Big) + \frac{4}{3}\nabla_{[\alpha}K_{\mu]\lambda}R^\lambda{}_{\nu\beta\rho}$$
$$+\Big(\frac{4}{3}\nabla_\sigma K_{\mu\lambda}R^\lambda{}_\nu{}^\sigma{}_\rho + \frac{2}{3}\nabla_\mu K_{\sigma\lambda}R^\lambda{}_\nu{}^\sigma{}_\rho\Big)g_{\alpha\beta} + \frac{16}{3}\nabla_\lambda\mathcal{Z}g_{\mu[\nu}{}^*R^\lambda{}_{\alpha]\beta\rho} - \frac{8}{3}\nabla_\mu\mathcal{Z}{}^*R_{\nu\alpha\beta\rho}\bigg]s^\alpha s^\beta\hat{p}^\mu\hat{p}^\nu\hat{p}^\rho.$$
(A.21)

The quantity into brackets appearing in the second line of this expression is actually vanishing:

$$\Big(\frac{4}{3}\nabla_\sigma K_{\mu\lambda}R^\lambda{}_\nu{}^\sigma{}_\rho + \frac{2}{3}\nabla_\mu K_{\sigma\lambda}R^\lambda{}_\nu{}^\sigma{}_\rho\Big)\hat{p}^\nu\hat{p}^\rho = \frac{2}{3}\big(2\nabla_\sigma K_{\mu\lambda} + \nabla_\mu K_{\sigma\lambda}\big)R^\lambda{}_\nu{}^\sigma{}_\rho\hat{p}^\nu\hat{p}^\rho$$
(A.22a)

$$= \frac{2}{3}\big(\nabla_\sigma K_{\mu\lambda} + \nabla_\lambda K_{\sigma\mu} + \nabla_\mu K_{\lambda\sigma}\big)R^\lambda{}_\nu{}^\sigma{}_\rho\hat{p}^\nu\hat{p}^\rho = 0,$$
(A.22b)

which follows from the definition of Killing tensors.

## A.3  Reduced expression

We can now write down the complete expression $\dot{\mathcal{Q}}^{(2)} = \dot{\mathcal{Q}}^{MD}_R + \dot{\mathcal{Q}}^Q_R + \dot{\mathcal{Q}}^{\text{quad}}$. Gathering all the pieces (A.3), (A.10), (A.21), we get

$$\dot{\mathcal{Q}}^{(2)} = \bigg[\kappa\nabla_\alpha\big(K_{\mu\lambda}R^\lambda{}_{\nu\beta\rho}\big) + \nabla_\nu\big(4N_{\alpha\mu\beta\rho} - \kappa K_{\mu\lambda}R^\lambda{}_{\alpha\beta\rho}\big) - \kappa\nabla_\mu K_{\nu\lambda}R^\lambda{}_{\alpha\beta\rho} + \frac{2\kappa}{3}\nabla_{[\mu}K_{\lambda]\alpha}R^\lambda{}_{\nu\beta\rho}$$
$$+ 2\big(Y_{\lambda\mu}\xi_\alpha - Y_{\lambda\alpha}\xi_\mu\big)R^{*\lambda}{}_{\nu\beta\rho} - \frac{16\kappa}{3}\nabla_\lambda\mathcal{Z}g_{\mu[\nu}R^{*\lambda}{}_{\alpha]\beta\rho} + 2\Big(\frac{4\kappa}{3} - 1\Big)\nabla_\mu\mathcal{Z}R^*_{\nu\alpha\beta\rho}\bigg]s^\alpha s^\beta\hat{p}^\mu\hat{p}^\nu\hat{p}^\rho$$
$$+ \mathcal{O}(\mathcal{S}^3).$$
(A.23)

In order to further simplify this expression we first derive some additional useful identities. One has

$$\nabla_\mu K_{\nu\lambda} R^\lambda{}_{\alpha\beta\rho} \hat{p}^\mu \hat{p}^\nu = Y^\sigma{}_\mu \nabla_\lambda Y_{\sigma\nu} R^\lambda{}_{\alpha\beta\rho} \hat{p}^\mu \hat{p}^\nu. \tag{A.24}$$

We obtain the relation

$$\nabla_\mu K_{\nu\lambda} R^\lambda{}_{\alpha\beta\rho} \hat{p}^\mu \hat{p}^\nu = \left[ \left( Y_{\lambda\mu}\xi_\alpha - Y_{\alpha\mu}\xi_\lambda \right) {}^*R^\lambda{}_{\nu\beta\rho} - g_{\alpha\nu} Y_{\lambda\mu}\xi_\kappa {}^*R^{\lambda\kappa}{}_{\beta\rho} \right] \hat{p}^\mu \hat{p}^\nu. \tag{A.25}$$

In a similar fashion, one can prove that

$$\nabla_{[\mu} K_{\lambda]\alpha} R^\lambda{}_{\nu\beta\rho} \hat{p}^\mu \hat{p}^\nu = \left[ \frac{3}{2} \left( Y_{\alpha\mu}\xi_\lambda + Y_{\lambda\alpha}\xi_\mu \right) {}^*R^\lambda{}_{\nu\beta\rho} - \frac{3}{2} g_{\mu\nu} Y_{\lambda\alpha}\xi_\kappa {}^*R^{\lambda\kappa}{}_{\beta\rho} \right.$$
$$\left. + \frac{1}{2} g_{\alpha\mu} \nabla_\lambda \mathcal{Z} {}^*R^\lambda{}_{\nu\beta\rho} - \frac{1}{2} g_{\mu\nu} \nabla_\lambda \mathcal{Z} {}^*R^\lambda{}_{\alpha\beta\rho} - \frac{1}{2} \nabla_\mu \mathcal{Z} {}^*R_{\alpha\nu\beta\rho} \right] \hat{p}^\mu \hat{p}^\nu. \tag{A.26}$$

When the dust settles, we are left with

$$\dot{\mathcal{Q}}^{(2)} = \left[ 4 \nabla_\mu N_{\alpha\nu\beta\rho} + 2\kappa \nabla_{[\alpha} \mathcal{M}^{(1)}_{|\mu|\nu]\beta\rho} + \kappa \left( g_{\alpha\mu} Y_{\lambda\nu} - g_{\mu\nu} Y_{\lambda\alpha} \right) \xi_\kappa {}^*R^{\lambda\kappa}{}_{\beta\rho} \right.$$
$$+ \left( 2\kappa Y_{\alpha\mu}\xi_\lambda + (2-\kappa)\left( Y_{\lambda\mu}\xi_\alpha + Y_{\alpha\lambda}\xi_\mu \right) + 3\kappa g_{\alpha\mu} \nabla_\lambda \mathcal{Z} \right) {}^*R^\lambda{}_{\nu\beta\rho} - 3\kappa g_{\mu\nu} \nabla_\lambda \mathcal{Z} {}^*R^\lambda{}_{\alpha\beta\rho}$$
$$\left. + (3\kappa - 2) \nabla_\mu \mathcal{Z} R^*_{\nu\alpha\beta\rho} \right] s^\alpha s^\beta \hat{p}^\mu \hat{p}^\nu \hat{p}^\rho + \mathcal{O}(\mathcal{S}^3), \tag{A.27}$$

where $\mathcal{M}^{(1)}_{\alpha\beta\gamma\delta} \triangleq K_{\alpha\lambda} R^\lambda{}_{\beta\gamma\delta}$. This precisely yields the constraint equation (54).

# B  Reducing the constraints with the covariant building blocks: Intermediate algebra

## B.1  Some identities

A bit of cumbersome (but straightforward) algebra leads to the following identities written most shortly in the $\alpha$-$\omega$ formulation,

$$Y_{\alpha\mu} s^\alpha \hat{p}^\mu = -\alpha_B^{(0,-1)}, \tag{B.1a}$$

$$\nabla_\mu \mathcal{Z} \hat{p}^\mu = -\alpha_A^{(0,-1)}, \tag{B.1b}$$

$$R^*_{\nu\alpha\beta\rho} s^\alpha s^\beta \hat{p}^\nu \hat{p}^\rho = 3M \omega_{B^2}^{(0,3)} + M \left( \mathcal{A}^2 + \mathcal{P}^2 \mathcal{S}^2 \right) \omega_1^{(0,3)}, \tag{B.1c}$$

$$\nabla_\lambda \mathcal{Z} R^{*\lambda}{}_{\alpha\beta\rho} s^\alpha s^\beta \hat{p}^\rho = \frac{3M}{2} \left( E_s \omega_B^{(0,2)} - D \omega_B^{(1,3)} \right) + M \left( \mathcal{S}^2 \alpha_A^{(0,-1)} - \mathcal{A} \alpha_C^{(0,-1)} \right) \omega_1^{(0,3)}, \tag{B.1d}$$

$$\nabla_\lambda \mathcal{Z} R^{*\lambda}{}_{\nu\beta\rho} s^\beta \hat{p}^\nu \hat{p}^\rho = \frac{3M}{2} \left( E \omega_B^{(0,2)} - F \omega_B^{(1,3)} \right) + M \left( \mathcal{A} \alpha_A^{(0,-1)} + \mathcal{P}^2 \alpha_C^{(0,-1)} \right) \alpha_1^{(0,3)}, \tag{B.1e}$$

$$Y_{\alpha\lambda} R^{*\lambda}{}_{\nu\beta\rho} s^\alpha s^\beta \hat{p}^\nu \hat{p}^\rho = -M \mathcal{A} \omega_B^{(0,2)} + \frac{M}{2} \left( \mathcal{A} \omega_{\bar{B}}^{(1,3)} - 3G \omega_B^{(1,3)} \right), \tag{B.1f}$$

$$Y_{\mu\lambda} R^{*\lambda}{}_{\nu\beta\rho} s^\beta \hat{p}^\mu \hat{p}^\nu \hat{p}^\rho = M \mathcal{P}^2 \omega_B^{(0,2)} - \frac{M}{2} \left( \mathcal{P}^2 \omega_{\bar{B}}^{(1,3)} + 3H \omega_B^{(1,3)} \right), \tag{B.1g}$$

$$Y_{\mu\lambda} R^{*\lambda}{}_{\alpha\beta\rho} s^\alpha s^\beta \hat{p}^\mu \hat{p}^\rho = -M \mathcal{A} \omega_B^{(0,2)} + \frac{M}{2} \left( \mathcal{A} \omega_{\bar{B}}^{(1,3)} - 3G \omega_B^{(1,3)} \right), \tag{B.1h}$$

$$\xi_\lambda R^{*\lambda}{}_{\nu\beta\rho} s^\beta \hat{p}^\nu \hat{p}^\rho = -3M \omega_{AB}^{(0,3)} + M \left( E_s \mathcal{P}^2 + E \mathcal{A} \right) \omega_1^{(0,3)}, \tag{B.1i}$$

$$Y_{\lambda\kappa} R^{*\lambda\kappa}{}_{\beta\rho} s^\beta \hat{p}^\rho = 4M \omega_B^{(0,2)}, \tag{B.1j}$$

$$Y_{\alpha\lambda}\xi_{\kappa}R^{*\lambda\kappa}{}_{\beta\rho}s^{\alpha}s^{\beta}\hat{p}^{\rho} = \frac{M}{2}\left(E_S\omega_B^{(0,2)} - 3D\omega_B^{(1,3)}\right) + M\left(\mathcal{A}\omega_C^{(0,-1)} - \mathcal{S}^2\omega_A^{(0,-1)}\right)\alpha_1^{(0,3)}, \quad \text{(B.1k)}$$

$$Y_{\nu\lambda}\xi_{\kappa}R^{*\lambda\kappa}{}_{\beta\rho}s^{\beta}\hat{p}^{\nu}\hat{p}^{\rho} = \frac{M}{2}\left(E\omega_B^{(0,2)} - 3F\omega_B^{(1,3)}\right) - M\left(\mathcal{A}\omega_A^{(0,-1)} + \mathcal{P}^2\omega_C^{(0,-1)}\right)\alpha_1^{(0,3)}, \quad \text{(B.1l)}$$

$$\xi_{\lambda}Y_{\alpha\kappa}R^{*\lambda}{}_{\rho\beta}{}^{\kappa}s^{\alpha}s^{\beta}\hat{p}^{\rho} = M\left(\mathcal{S}^2\omega_A^{(0,2)} + E_s\omega_{\bar{B}}^{(1,3)} - \mathcal{A}\omega_{\bar{C}}^{(1,3)}\right) + \frac{M}{2}\left(3I\omega_A^{(1,3)} + \mathcal{S}^2\omega_{\bar{A}}^{(1,3)}\right), \quad \text{(B.1m)}$$

$$\xi_{\lambda}Y_{\beta\kappa}R^{*\lambda}{}_{\nu\rho}{}^{\kappa}s^{\beta}\hat{p}^{\nu}\hat{p}^{\rho} = \frac{3M}{2}\left(\mathcal{A}\omega_A^{(0,2)} + G\omega_A^{(1,3)}\right) + M\left(E\alpha_B^{(0,-1)} + \mathcal{P}^2\alpha_C^{(0,-1)}\right)\omega_1^{(0,3)}. \quad \text{(B.1n)}$$

Let us turn to identities involving covariant derivatives of the Riemann tensor. Making use of the identities (67) enforces the fundamental relation:

$$R_{\alpha\nu\beta\rho;\mu} = M\nabla_{\mu}\text{Re}\left(\frac{3(\mathcal{R}N_{\alpha\nu})(\mathcal{R})N_{\beta\rho} - \mathcal{R}^2 G_{\alpha\nu\beta\rho}}{\mathcal{R}^5}\right)$$

$$= -M\,\text{Im}\left\{\mathcal{R}^{-4}\left[5N_{\mu\lambda}\xi^{\lambda}\left(3N_{\alpha\nu}N_{\beta\rho} - G_{\alpha\nu\beta\rho}\right) - 3\left(G_{\alpha\nu\mu\lambda}\xi^{\lambda}N_{\beta\rho} + N_{\alpha\nu}G_{\beta\rho\mu\lambda}\xi^{\lambda}\right)\right.\right.$$

$$\left.\left. + 2N_{\mu\lambda}\xi^{\lambda}G_{\alpha\nu\beta\rho}\right]\right\}. \quad \text{(B.2)}$$

It leads to the following 'differential-to-algebraic' identities:

$$\nabla_{\mu}R_{\alpha\nu\beta\rho}s^{\alpha}s^{\beta}\hat{p}^{\mu}\hat{p}^{\nu}\hat{p}^{\rho} = 3M\,\text{Im}\left\{\mathcal{R}^{-4}\left[5AB^2 - 2B\left(\mathcal{A}E + \mathcal{P}^2 E_S\right) + A\left(\mathcal{S}^2\mathcal{P}^2 + \mathcal{A}^2\right)\right]\right\}, \quad \text{(B.3a)}$$

$$K_{\alpha\lambda}\nabla_{\mu}R^{\lambda}{}_{\nu\beta\rho}s^{\alpha}s^{\beta}\hat{p}^{\mu}\hat{p}^{\nu}\hat{p}^{\rho} = -\frac{3M}{2}\text{Re}\left(\mathcal{R}^2\right)\text{Im}\left\{\mathcal{R}^{-4}\left[5AB^2 - 2B\left(\mathcal{A}E + \mathcal{P}^2 E_S\right) + A\left(\mathcal{S}^2\mathcal{P}^2 + \mathcal{A}^2\right)\right]\right\}$$

$$-\frac{3M}{2}|\mathcal{R}|^2\,\text{Im}\left\{\mathcal{R}^{-4}\left[5A|B|^2 + A(\mathcal{P}^2 I + \mathcal{A}G) - B\left(GE - D\mathcal{P}^2\right)\right.\right.$$

$$\left.\left. - \bar{B}\left(\mathcal{A}E + \mathcal{P}^2 E_s\right)\right]\right\}, \quad \text{(B.3b)}$$

$$K_{\nu\lambda}\nabla_{\mu}R^{\lambda}{}_{\alpha\beta\rho}s^{\alpha}s^{\beta}\hat{p}^{\mu}\hat{p}^{\nu}\hat{p}^{\rho} = \frac{3M}{2}\text{Re}\left(\mathcal{R}^2\right)\text{Im}\left\{\mathcal{R}^{-4}\left[5AB^2 - 2B\left(\mathcal{A}E + \mathcal{P}^2 E_S\right) + A\left(\mathcal{S}^2\mathcal{P}^2 + \mathcal{A}^2\right)\right]\right\}$$

$$+\frac{3M}{2}|\mathcal{R}|^2\,\text{Im}\left\{\mathcal{R}^{-4}\left[5A|B|^2 + A\left(\mathcal{A}G - \mathcal{S}^2 H\right)\right.\right.$$

$$\left.\left. + B\left(E_s H + \mathcal{A}F + \bar{A}B - A\bar{B}\right) - \bar{B}\left(\mathcal{A}E + \mathcal{P}^2 E_s\right)\right]\right\}, \quad \text{(B.3c)}$$

$$K_{\mu\lambda}\nabla_{\alpha}R^{\lambda}{}_{\nu\beta\rho}s^{\alpha}s^{\beta}\hat{p}^{\mu}\hat{p}^{\nu}\hat{p}^{\rho} = \frac{3M}{2}|\mathcal{R}|^2\,\text{Im}\left\{\mathcal{R}^{-4}\left[C\left(\mathcal{P}^2 G + \mathcal{A}H\right)\right.\right.$$

$$\left.\left. - B(EG + \mathcal{A}F) + B\left(\bar{A}B - A\bar{B}\right)\right]\right\}. \quad \text{(B.3d)}$$

## B.2 Ansatz terms for the quadratic invariant.

We provide here the explicit form of the terms constituting the Ansatz for the black hole quadratic invariant in terms of the covariant building blocks. We use the notation $N^{(A)} \triangleq N^{(A)}_{\mu\alpha\nu\beta}s^{\alpha}s^{\beta}\hat{p}^{\mu}\hat{p}^{\nu}$ $(A = 1,\ldots,4)$.

$$N^{(1)} = -M|B|^2\alpha_1^{(1,2)} + \frac{M}{4}\left[3\left(\alpha_{B^2}^{(0,1)} + \alpha_{B^2}^{(2,3)}\right) + \left(\mathcal{A}^2 + \mathcal{P}^2\mathcal{S}^2\right)\left(\alpha_1^{(0,1)} + \alpha_1^{(2,3)}\right)\right], \quad \text{(B.4a)}$$

$$N^{(2)} = -\frac{M}{4}\left[\left(\mathcal{A}^2 + \mathcal{P}^2\mathcal{S}^2\right)\alpha_1^{(0,1)} + \alpha_{B^2}^{(0,1)}\right], \quad \text{(B.4b)}$$

$$N^{(3)} = \frac{1}{4}\left[-\left(\mathcal{A}^2 + \mathcal{P}^2\mathcal{S}^2\right) + E\left(E\mathcal{S}^2 - E_S\mathcal{A}\right) - E_S\left(E_S\mathcal{P}^2 + \mathcal{A}E\right)\right] + \frac{M}{2}\left(\mathcal{A}^2 + \mathcal{P}^2\mathcal{S}^2\right)\alpha_1^{(0,1)}, \quad \text{(B.4c)}$$

$$N^{(4)} = \frac{1}{2}\left(\mathcal{A}^2 + \mathcal{P}^2\mathcal{S}^2\right)\left(2M\alpha_1^{(0,1)} - 1\right). \quad \text{(B.4d)}$$

## B.3 Derivatives of the scalar basis.

A direct computation gives the following identities

$$\hat{\nabla}\mathcal{S}^2 = \hat{\nabla}\mathcal{P}^2 = \hat{\nabla}\mathcal{A} = 0, \tag{B.5a}$$

$$\hat{\nabla}A = -i\big(E^2 + \mathcal{P}^2\xi^2\big)\mathcal{R}^{-1} + \frac{iM}{2}\bigg(\frac{\mathcal{P}^2}{\mathcal{R}^2} + \frac{H}{\bar{\mathcal{R}}^2}\bigg) - iA^2\mathcal{R}^{-1}, \tag{B.5b}$$

$$\hat{\nabla}B = i\big(\mathcal{A}E + \mathcal{P}^2 E_s\big)\mathcal{R}^{-1} - iAB\mathcal{R}^{-1}, \tag{B.5c}$$

$$\hat{\nabla}C = i\big(\mathcal{A}\xi^2 - EE_s\big)\mathcal{R}^{-1} + \frac{iM}{2}\bigg(\frac{G}{\bar{\mathcal{R}}^2} - \frac{\mathcal{A}}{\mathcal{R}^2}\bigg) - iAC\mathcal{R}^{-1}, \tag{B.5d}$$

$$\hat{\nabla}D = 2E\omega_{\bar{C}}^{(0,1)} - 2\xi^2\omega_{\bar{B}}^{(0,1)} + M\omega_B^{(0,2)} + 2D\omega_A^{(0,1)}, \tag{B.5e}$$

$$\hat{\nabla}E = 0, \tag{B.5f}$$

$$\hat{\nabla}E_s = -M\omega_B^{(0,2)}, \tag{B.5g}$$

$$\hat{\nabla}F = 2E\omega_{\bar{A}}^{(0,1)} + 2F\omega_A^{(0,1)}, \tag{B.5h}$$

$$\hat{\nabla}G = 2G\omega_A^{(0,1)} + 2E\omega_{\bar{B}}^{(0,1)} + 2\mathcal{P}^2\omega_{\bar{C}}^{(0,1)}, \tag{B.5i}$$

$$\hat{\nabla}H = 2\omega_A^{(0,1)}H + 2\mathcal{P}^2\omega_{\bar{A}}^{(0,1)}, \tag{B.5j}$$

$$\hat{\nabla}I = 2\mathcal{S}^2\omega_{\bar{A}}^{(0,1)} + 4E_s\omega_{\bar{B}}^{(0,1)} - 4\mathcal{A}\omega_{\bar{C}}^{(0,1)} + 2I\omega_A^{(0,1)}. \tag{B.5k}$$

## B.4 Directional derivatives of the Ansatz terms

We aim to compute the contributions $\hat{\nabla}N^{(A)}$. Let us proceed step by step. First, we compute $N_{\alpha\beta\gamma\delta}^{(A)}$ for each $A = 1, \dots 4$. In Ricci-flat spacetimes, using the identity (A.16), we obtain

$$N_{\alpha\beta\gamma\delta}^{(1)} = -\frac{1}{2}KR_{\alpha\beta\gamma\delta} + \mathcal{M}_{[\alpha\beta]\gamma\delta}^{(1)}, \tag{B.6}$$

where $K \triangleq K^\alpha{}_\alpha$. Moreover, noticing that

$$^*\mathcal{M}_{\alpha\beta\gamma\delta}^{(2)} = Y_{\lambda[\alpha} {}^*R^\lambda{}_{\beta]\sigma\delta}Y^\sigma{}_\gamma = -R^*{}_{\sigma\delta[\alpha}{}^\lambda Y_{\beta]\lambda}Y^\sigma{}_\gamma, \tag{B.7}$$

and using the symmetries of the Riemann tensor, we can write

$$N_{\alpha\nu\beta\rho}^{(2)} = -Y_{\lambda[\alpha} {}^*R^*{}^\lambda{}_{\nu][\beta}{}^\sigma Y_{\rho]\sigma}. \tag{B.8}$$

In Ricci-flat spacetimes, this boils down to

$$N_{\alpha\nu\beta\rho}^{(2)} = Y_{\lambda[\alpha}R^\lambda{}_{\nu][\beta}{}^\sigma Y_{\rho]\sigma}. \tag{B.9}$$

The two last computations are more straightforward and give

$$N_{\alpha\beta\gamma\delta}^{(3)} = \frac{1}{2}N_{\alpha\beta\gamma\delta}^{(4)} - \xi_{[\alpha}g_{\beta][\gamma}\xi_{\delta]}, \qquad N_{\alpha\beta\gamma\delta}^{(4)} = -g_{\alpha[\gamma}g_{\delta]\beta}\xi^2. \tag{B.10}$$

We shall evaluate the following covariant derivatives:

$$\nabla_\mu\big(KR_{\alpha\nu\beta\rho}\big)s^\alpha s^\beta \hat{p}^\mu \hat{p}^\nu \hat{p}^\rho = \big(\nabla_\mu K R_{\alpha\nu\beta\rho} + K\nabla_\mu R_{\alpha\nu\beta\rho}\big)s^\alpha s^\beta \hat{p}^\mu \hat{p}^\nu \hat{p}^\rho. \tag{B.11}$$

Using (107) the first term of the right-hand side of (B.11) can be written

$$
\nabla_\mu K R_{\alpha\nu\beta\rho} s^\alpha s^\beta \hat{p}^\mu \hat{p}^\nu \hat{p}^\rho = \Big\{ 4\big(\xi_\lambda Y_{\alpha\mu} + \xi_\mu Y_{\lambda\alpha} - \xi_\alpha Y_{\lambda\mu}\big) R^{*\lambda}{}_{\nu\beta\rho}
$$

$$
+ 2\big[ g_{\alpha\mu}(2Y_{\lambda\nu}\xi_\kappa - Y_{\lambda\kappa}\xi_\nu) + g_{\mu\nu}(2\xi_\lambda Y_{\kappa\alpha} + \xi_\alpha Y_{\lambda\kappa})\big] R^{*\lambda\kappa}{}_{\beta\rho} \Big\} s^\alpha s^\beta \hat{p}^\mu \hat{p}^\nu \hat{p}^\rho \tag{B.12a}
$$

$$
= 4\Big[ \big(\xi_\lambda Y_{\alpha\mu} + \xi_\mu Y_{\lambda\alpha} - \xi_\alpha Y_{\lambda\mu}\big) R^{*\lambda}{}_{\nu\beta\rho} + \big(Y_{\lambda\kappa}\xi_{[\alpha} + 2\xi_\lambda Y_{\kappa[\alpha}\big) g_{\mu]\nu} R^{*\lambda\kappa}{}_{\beta\rho} \Big] s^\alpha s^\beta \hat{p}^\mu \hat{p}^\nu \hat{p}^\rho \,. \tag{B.12b}
$$

Gathering the two pieces above yields

$$
\nabla_\mu N^{(1)}_{\alpha\nu\beta\rho} s^\alpha s^\beta \hat{p}^\mu \hat{p}^\nu \hat{p}^\rho = \Big[ K_{\lambda[\alpha} R^\lambda{}_{\nu]\beta\rho;\mu} - \frac{1}{2} K \nabla_\mu R_{\alpha\nu\beta\rho} - \frac{1}{2} \nabla_\mu \mathcal{Z} R^*_{\alpha\nu\beta\rho}
$$

$$
- \frac{1}{2}\big(\nabla_\lambda \mathcal{Z} g_{\mu\nu} + Y_{\lambda\mu}\xi_\nu\big) R^{*\lambda}{}_{\alpha\beta\rho} + \frac{1}{2}\big(\nabla_\lambda \mathcal{Z} g_{\mu\alpha} + 3Y_{\lambda\mu}\xi_\alpha + Y_{\mu\alpha}\xi_\lambda
$$

$$
+ 2Y_{\alpha\lambda}\xi_\mu\big) R^{*\lambda}{}_{\nu\beta\rho} - 2\big(Y_{\lambda\kappa}\xi_{[\alpha} + \xi_\lambda Y_{\kappa[\alpha}\big) g_{\mu]\nu} R^{*\lambda\kappa}{}_{\beta\rho} \Big] s^\alpha s^\beta \hat{p}^\mu \hat{p}^\nu \hat{p}^\rho \,. \tag{B.13}
$$

On the other hand,

$$
\nabla_\mu N^{(2)}_{\alpha\nu\beta\rho} s^\alpha s^\beta \hat{p}^\mu \hat{p}^\nu \hat{p}^\rho = \nabla_\mu \Big( Y_{\lambda[\alpha} R^\lambda{}_{\nu][\beta}{}^\sigma Y_{\rho]\sigma} \Big) s^\alpha s^\beta \hat{p}^\mu \hat{p}^\nu \hat{p}^\rho \tag{B.14a}
$$

$$
= \nabla_\mu \Big( Y_{\lambda\alpha} R^\lambda{}_{\nu\beta}{}^\sigma Y_{\rho\sigma} \Big) \hat{p}^\mu s^{[\alpha} \hat{p}^{\nu]} s^{[\beta} \hat{p}^{\rho]} \tag{B.14b}
$$

$$
= \Big( 2\nabla_\mu Y_{\lambda\alpha} Y_{\rho\sigma} R^\lambda{}_{\nu\beta}{}^\sigma + Y_{\lambda\alpha} Y_{\rho\sigma} \nabla_\mu R^\lambda{}_{\nu\beta}{}^\sigma \Big) \hat{p}^\mu s^{[\alpha} \hat{p}^{\nu]} s^{[\beta} \hat{p}^{\rho]} \tag{B.14c}
$$

$$
= \Big( 2\varepsilon_{\mu\lambda\alpha\kappa} \xi^\kappa Y_{\rho\sigma} R^\lambda{}_{\nu\beta}{}^\sigma + Y_{\lambda\alpha} Y_{\rho\sigma} \nabla_\mu R^\lambda{}_{\nu\beta}{}^\sigma \Big) \hat{p}^\mu s^{[\alpha} \hat{p}^{\nu]} s^{[\beta} \hat{p}^{\rho]} \tag{B.14d}
$$

$$
= \frac{1}{2} 2\varepsilon_{\mu\lambda\alpha\kappa} \xi^\kappa Y_{\rho\sigma} R^\lambda{}_{\nu\beta}{}^\sigma s^\alpha \hat{p}^\mu \hat{p}^\nu s^{[\beta} \hat{p}^{\rho]} + Y_{\lambda[\alpha|} \nabla_\mu R^\lambda{}_{|\nu][\beta}{}^\sigma Y_{\rho]\sigma} s^\alpha s^\beta \hat{p}^\mu \hat{p}^\nu \hat{p}^\rho \tag{B.14e}
$$

$$
= \Big[ \frac{1}{2} Y_{\alpha\lambda}\xi_\mu R^{*\lambda}{}_{\nu\beta\rho} - \frac{1}{2} Y_{\mu\lambda}\xi_\nu R^{*\lambda}{}_{\alpha\beta\rho} - \frac{1}{2}\big(g_{\mu\nu} Y_{\alpha\kappa} + g_{\alpha\mu} Y_{\nu\kappa}\big)\xi_\lambda R^{*\lambda}{}_{\rho\beta}{}^\kappa
$$

$$
+ \frac{1}{2} g_{\mu\nu}\xi_\lambda Y_{\rho\kappa} R^{*\lambda}{}_{\alpha\beta}{}^\kappa + \frac{1}{2} g_{\alpha\mu}\xi_\lambda Y_{\beta\kappa} R^{*\lambda}{}_{\nu\rho}{}^\kappa + Y_{\lambda[\alpha|} \nabla_\mu R^\lambda{}_{|\nu][\beta}{}^\sigma Y_{\rho]\sigma} \Big] s^\alpha s^\beta \hat{p}^\mu \hat{p}^\nu \hat{p}^\rho \,. \tag{B.14f}
$$

Finally, one has

$$
\nabla_\mu \big(\xi_{(\alpha}\xi_{\beta)}\big) = 2\nabla_\mu \xi_{(\alpha}\xi_{\beta)} \,. \tag{B.15}
$$

In Ricci-flat spacetimes, Eq. (157) of [25] boils down to the identity

$$
\nabla_\alpha \xi_\beta = -\frac{1}{4} R^*_{\alpha\beta\gamma\delta} Y^{\gamma\delta} \,. \tag{B.16}
$$

All in all, we obtain the relations

$$
\nabla_\mu \big(\xi_{(\alpha}\xi_{\beta)}\big) = \frac{1}{2}\xi_{(\alpha} R^*_{\beta)\mu\gamma\delta} Y^{\gamma\delta} \,, \qquad \nabla_\mu \big(\xi^2\big) = \frac{1}{2}\xi_\lambda R^{*\lambda}{}_{\mu\gamma\delta} Y^{\gamma\delta} \,. \tag{B.17}
$$

This yields

$$\nabla_\mu N^{(3)}_{\alpha\nu\beta\rho} s^\alpha s^\beta \hat{p}^\mu \hat{p}^\nu \hat{p}^\rho = \left[\frac{1}{2}\nabla_\mu N^{(4)}_{\alpha\nu\beta\rho} - \nabla_\mu\left(\xi_{[\alpha}g_{\nu][\beta}\xi_{\rho]}\right)\right] s^\alpha s^\beta \hat{p}^\mu \hat{p}^\nu \hat{p}^\rho \tag{B.18a}$$

$$= \left[\frac{1}{2}\nabla_\mu N^{(4)}_{\alpha\nu\beta\rho} + \frac{1}{2}\xi_{[\alpha}g_{\nu][\rho}R^*_{\beta]\mu\gamma\delta}Y^{\gamma\delta}\right] s^\alpha s^\beta \hat{p}^\mu \hat{p}^\nu \hat{p}^\rho \tag{B.18b}$$

$$= \left[\frac{1}{2}\nabla_\mu N^{(4)}_{\alpha\nu\beta\rho} + \frac{1}{4}\xi_{[\alpha}g_{\mu]\nu}R^*_{\beta\rho\gamma\delta}Y^{\gamma\delta}\right] s^\alpha s^\beta \hat{p}^\mu \hat{p}^\nu \hat{p}^\rho , \tag{B.18c}$$

as well as

$$\nabla_\mu N^{(4)}_{\alpha\nu\beta\rho} s^\alpha s^\beta \hat{p}^\mu \hat{p}^\nu \hat{p}^\rho = -g_{\alpha[\beta}g_{\rho]\nu}\nabla_\mu(\xi^2) s^\alpha s^\beta \hat{p}^\mu \hat{p}^\nu \hat{p}^\rho$$

$$= -\frac{1}{2}g_{\alpha[\beta}g_{\mu]\nu}\xi_\lambda R^{*\lambda}{}_{\rho\gamma\delta}Y^{\gamma\delta} s^\alpha s^\beta \hat{p}^\mu \hat{p}^\nu \hat{p}^\rho . \tag{B.19}$$

Let us now use the preceding relations to write down the desired contribution in the covariant building blocks language. We will demonstrate the procedure on the $A = 1$ term, which turns out to be the most involved to compute. The computations of the others contributions will not be detailed in this text. One has

$$\left(K_{\lambda[\alpha}R^\lambda{}_{\nu]\beta\rho;\mu} - \frac{1}{2}K\nabla_\mu R_{\alpha\nu\beta\rho}\right)s^\alpha s^\beta \hat{p}^\mu \hat{p}^\nu \hat{p}^\rho$$

$$= \frac{15M}{4}\left(\omega^{(0,2)}_{AB^2} + \omega^{(2,4)}_{AB^2}\right) + \frac{3M}{4}\left(\mathcal{S}^2\mathcal{P}^2 + \mathcal{A}^2\right)\left(\omega^{(0,2)}_A + \omega^{(2,4)}_A\right)$$

$$- \frac{3M}{2}\left(\mathcal{A}E + \mathcal{P}^2 E_s\right)\left(\omega^{(0,2)}_B + \omega^{(2,4)}_B\right) - \frac{3M}{4}\left(\mathcal{P}^2 I - \mathcal{S}^2 H + 10|B|^2 + 2\mathcal{A}G\right)\omega^{(1,3)}_A$$

$$- \frac{3M}{4}\left(\mathcal{P}^2 D - EG + E_s H + \mathcal{A}F\right)\omega^{(1,3)}_B + \frac{3M}{2}\left(\mathcal{A}E + \mathcal{P}^2 E_s\right)\omega^{(1,3)}_{\bar{B}} - \frac{3M}{2}\text{Im}\left(\bar{A}B\right)\alpha^{(1,3)}_B . \tag{B.20}$$

Using the various identities derived above in Eq. (B.13) allows to write

$$\hat{\nabla}N^{(1)} = \frac{15M}{4}\left(\omega^{(0,2)}_{AB^2} + \omega^{(2,4)}_{AB^2}\right) - \frac{3M}{2}\left(\omega^{(0,3)}_{B^2}\alpha^{(0,-1)}_A + \omega^{(0,3)}_{AB}\alpha^{(0,-1)}_B\right)$$

$$+ \frac{M}{4}\left(\mathcal{A}^2 + \mathcal{P}^2\mathcal{S}^2\right)\left(\omega^{(0,2)}_A + 2\omega^{(1,3)}_{\bar{A}} + 3\omega^{(2,4)}_A\right)$$

$$- \frac{M}{2}\left(\mathcal{A}E + \mathcal{P}^2 E_s\right)\left(5\omega^{(0,2)}_B - 2\omega^{(1,3)}_{\bar{B}} + 3\omega^{(2,4)}_B\right)$$

$$- \frac{3M}{4}\left(10|B|^2 + 2\mathcal{A}G + \mathcal{P}^2 I - \mathcal{S}^2 H\right)\omega^{(1,3)}_A - 3M\left(\mathcal{A}F - EG + E_s H + \mathcal{P}^2 D\right)\omega^{(1,3)}_B$$

$$- \frac{3M}{2}\omega^{(0,0)}_{\bar{A}B}\alpha^{(1,3)}_B . \tag{B.21}$$

Making use of the relations (103) allows to express $DN^{(1)}$ in terms of linearly independent contributions. When the dust settles down, we are left with

$$\hat{\nabla}N^{(1)} = \frac{M}{4}\left(\mathcal{A}^2 + \mathcal{P}^2\mathcal{S}^2\right)\left(\omega^{(0,2)}_A + 2\omega^{(1,3)}_{\bar{A}} + 3\omega^{(2,4)}_A\right)$$

$$- \frac{M}{2}\left(\mathcal{A}E + \mathcal{P}^2 E_s\right)\left(5\omega^{(0,2)}_B - 2\omega^{(1,3)}_{\bar{B}} + 3\omega^{(2,4)}_B\right)$$

$$- \frac{9M}{2}|B|^2\omega^{(1,3)}_A - \frac{3M}{2}\left(\mathcal{A}F - EG + E_s H + \mathcal{P}^2 D\right)\omega^{(1,3)}_B$$

$$+ \frac{9M}{4}\omega^{(0,2)}_{AB^2} + \frac{15M}{4}\omega^{(2,4)}_{AB^2} . \tag{B.22}$$

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
