# Peer review of "Generalized Carter constant for quadrupolar test bodies in Kerr spacetime"

_SciPost Physics, doi:SciPost Phys. 15, 226 (2023)_

## Round 1 · Referee Report · Scott Hughes (Referee 1) · 2023-7-29

Strengths

Many! - see report for details.

Weaknesses

None

Report

Understanding the motion of a body under the influence of some potential is one of the oldest and most important problems in physics. The motion of a body in some prescribed spacetime is the relativistic generalization of this problem. Of particular interest is understanding whether, in various physically interesting circumstances, the motion is "integrable" or not. If it is, then it should be possible to make high-accuracy predictive models of a system's long term behavior; if it is not, then the motion is generally chaotic.

Over the years, multiple claims have been made that binary systems in general relativity will evolve chaotically. In particular, claims have been made that the motion of a spinning body in the spacetime of a rotating black hole is not integrable. These claims have generally not held up under close scrutiny: they typically involve approximate equations of motion integrated beyond their realm of validity, or the motion of bodies with highly unphysical parameters. Indeed, over the years work has precisely clarified the limitations of this claim. Perhaps the most recent and important substantial advance is Witzany's proof that the motion of a spinning body is integrable if the equations of motion are truncated at leading order in the small body's spin. This proof left open the possibility that orbits may be chaotic if the equations of motion are not so truncated.

This outstanding and important paper advances this subject in two critical ways:

  • First, this paper proves that a spinning body's motion is in fact integrable at the next order in the small body's spin IF the body has the multipolar structure of a Kerr black hole. Key to the proof is to note that mass and spin are the two lowest multipoles in a tower of multipole moments with which a body can be endowed. If a body spins, then it could also have a quadrupole moment. This paper demonstrates that "order spin squared" terms associated with this quadrupole moment combine with quadratic order terms associated with the body's spin in such a way as to yield integrable motion, but only if the quadrupole moment has the "right" value. In particular, this work proves that the motion is integrable only if the body moving in a black hole spacetime has the multipolar structure of a black hole. Otherwise, chaotic evolution is indeed expected.

  • Second, this paper lays out in very clear language the framework by which this proof is developed, making it possible (at least in principle) to go beyond this proof. If one so wishes, one could follow this framework and examine whether the result continues to hold at higher orders. Though the analysis is dense and technical, everything is presented very clearly. In addition to presenting an important physical result, this paper serves as a technical manual for readers seeking to understand the integrability of orbits in some prescribed spacetime.

With great enthusiasm, I recommend this paper for publication. I have only three minor comments, as well as a larger-scale question which is perhaps fodder for the concluding discussion.

Minor comment 1: The text following Equation (51) discusses the linear-in-spin quantity $Q^{\rm lin}$, which the authors note has been extensively discussed in Ref. [25]. Perhaps I overlooked the relevant text, but $Q^{\rm lin}$ does not appear to have been defined in this paper. $Q^{\rm lin}$ appears to be a relabeling of certain pieces of the quantities which go into $Q^{(2)}$ [cf. Eq (49)]. Given how carefully everything is developed and defined in this paper, it would be useful to clarify this term as well.

Minor comment 2: In the text following Eq. (79), they note that the quantity $\mathcal{A}$ is "unphysical". I believe this is because the authors are using the Tulczyjew spin supplementary condition, and thus (chasing through the various definitions) we expect $\mathcal{A} = 0$ once all quantities take their physically allowed values. If I am reading this correctly, it might be useful to describe non-zero $\mathcal{A}$ as being unphysical under these circumstances.

Minor comment 3: It appears to me that there is a sign issue in Equations (92) - (95). Equation (92) shows that certain combinations of the Killing-Yano tensor contracted with derivatives of $\hat p_\alpha$ combine to give zero to quadratic order in the body's spin. Equations (93) and (94) show the values taken by those individual Killing-Yano / derivative contractions. When I insert Equations (93) and (94) as written into Equation (92), I do not find the claimed result: rather than leaving a result proportional to $\delta\kappa$, I find a right-hand side that is proportional to $\kappa + 1$. Perhaps my arithmetic has gone awry, but I get the result the authors claim (which is crucial for their proof) if some of the signs are flipped in Equations (92), (93), or (94).

Larger-scale question: This paper raises, to my mind, the possibility of an interesting bit of speculation (one which is in fact alluded to in the current concluding text). Suppose one continued the logic of this analysis to seek deformations of invariants of the motion to higher order in the multipole expansion. It seems plausible that we will find that such invariants exist and yield integrable motion provided that the orbiting body's multipole moments take the values prescribed by general relativity for a Kerr black hole. In other words, if the orbiting body has octupolar / sextadecapolar / etc moments set by its spin in accordance with the Kerr hypothesis, perhaps these invariants exist and the motion is integrable. It would perhaps be worthwhile to outline the program that would be necessary to investigate this speculation, which is consistent with the paper's current concluding text (particularly the line "One could investigate whether deformations ..." in the paper's concluding paragraph).

Requested changes

Included in the comments in the report. The only one which is really important, if I am correct, is the possibility of some sign flips as listed in "Minor comment 3".

---

## Round 1 · Referee Report · Anonymous (Referee 2) · 2023-10-18

Strengths

1- Main result is the exhibition of a constant of motion for spinning black holes (quadratic in spin) which is not present for neutron stars. This could be a useful handle in the future for distinguishing one from the other.

2- The paper offers a concise and thorough review of the dynamics of test bodies endowed with quadrupole in curved space and derivation of constraint equations.

3-Mathematica notebook as complementary material are provided.

Weaknesses

None

Report

In my view this paper paves the way for future investigations as not only obtained a new result, but introduced a new method to analyse orbit in Kerr space that could be generalized to higher multipoles. Moreover it gives lays down a systematic approach to the constrained dynamics problem of the spinning particle

Requested changes

None

---

## Round 2 · Referee Report · Scott Hughes (Referee 1) · 2023-11-1

Report

I am fully happy with the authors' responses to my report. I confess I have not read through the revised manuscript as thoroughly as I did for my initial report, but I read it closely enough to examine the changes they made.

I am happy to recommend acceptance with no further changes.

---

## Round 2 · Referee Report · Anonymous (Referee 2) · 2023-11-3

Report

I can recommend the paper for publication without further changes.

---

## Round 2 · Author Response

First, we would like to thank the referees for their very encouraging comments on our manuscript.

  • Regarding $Q^{\text{lin}}$, we meant that it was identical to R\”udiger’s invariant $Q^R$ but we omitted to state it. We have now simplified the notation and defined $Q^{\text{lin}}$ directly in Eqs. (49) and (50) and then explained that it is equal to R\”udiger’s $Q_R$, see arguments around Eq. (53).

  • Regarding $\mathcal A$, we agree with the comments of the referee. We added further comments on the significance of $\mathcal Q$ after Eq. (79).

  • Thanks to the referee for pointing out to a sign mistake. It has now been corrected. Please see the updated Eqs. (93), (94), (95). We also corrected a factor in Eqs. (100) and (101).

  • Regarding the existence of conserved quantities with higher multipolar content, we are totally in line with the referee. We added one further paragraph on these prospects in the conclusions following the suggestion of the referee.

---

## Editorial Decision

published